



# Mineral dust cycle in the Multiscale Online Nonhydrostatic AtmospheRe CHemistry model (MONARCH) Version 2.0

Martina Klose[1,10], Oriol Jorba[1], María Gonçalves Ageitos[1,2], Jeronimo Escribano[1], Matthew
L. Dawson[1,3], Vincenzo Obiso[1,4], Enza Di Tomaso[1], Sara Basart[1], Gilbert Montané Pinto[1],
Francesca Macchia[1], Paul Ginoux[5], Juan Guerschman[6], Catherine Prigent[7], Yue Huang[8], Jasper F. Kok[8],
Ron L. Miller[4], and Carlos Pérez García-Pando[1,9]

[1]Barcelona Supercomputing Center (BSC), Barcelona, Spain
[2]Technical University of Catalonia (UPC), Barcelona, Spain
[3]National Center for Atmospheric Research (NCAR), Boulder, CO, USA
[4]NASA Goddard Institute for Space Studies (GISS), New York, NY, USA
[5]Geophysical Fluid Dynamics Laboratory (GFDL), Princeton, NJ, USA
[6]Commonwealth Scientific and Industrial Research Organisation (CSIRO), Canberra, Australia
[7]Observatoire de Paris, PSL University, Sorbonne Université, CNRS, LERMA, Paris, France
[8]University of California, Los Angeles, CA, USA
[9]ICREA, Catalan Institution for Research and Advanced Studies, Barcelona, Spain
[10]present address: Karlsruhe Institute of Technology (KIT), Institute of Meteorology and Climate Research (IMK-TRO),
Department Troposphere Research, Karlsruhe, Germany

**Correspondence:** Martina Klose (martina.klose@kit.edu), Carlos Pérez García-Pando (carlos.perez@bsc.es)

**Abstract.** We present the dust module in the Multiscale Online Non-hydrostatic AtmospheRe CHemistry model (MONARCH) Version 2.0, a chemical weather prediction system that can be used for regional and global modeling at a range of resolutions. The representations of dust processes in MONARCH were upgraded with a focus on dust emission (emission parameterizations, entrainment thresholds, considerations of soil moisture and surface cover), lower boundary conditions (roughness, potential dust sources), and dust–radiation interactions. MONARCH now allows modeling of global and regional mineral dust cycles using fundamentally different paradigms, ranging from strongly simplified to physics-based parameterizations. We present a detailed description of these updates along with four global benchmark simulations, which use conceptually different dust emission parameterizations, and we evaluate the simulations against observations of dust optical depth. We determine key dust parameters, such as global annual emission/deposition flux, dust loading, dust optical depth, mass-extinction efficiency, single-scattering albedo, direct radiative effects. The total annual dust emission and deposition fluxes obtained with our four experiments, range between about 3,500 and 6,000 Tg, which largely depend upon differences in the emitted size distribution. Considering ellipsoidal particle shapes and dust refractive indices that account for size-resolved mineralogy, we estimate the global total (longwave and shortwave) dust direct radiative effect (DRE) at the surface to range between about $-0.90$ and $-0.63\,\mathrm{W\,m^{-2}}$ and at the top of the atmosphere between $-0.20$ and $-0.28\,\mathrm{W\,m^{-2}}$. Our evaluation demonstrates that MONARCH is able to reproduce key features of the spatio-temporal variability of the global dust cycle with important and insightful differences between the different configurations.



# 1 Introduction

The Multiscale Online Non-hydrostatic AtmospheRe CHemistry model (MONARCH) is a chemical weather modeling system
that can be used at multiple spatial scales, ranging from regional scales at single-digit kilometer resolutions with explicit con-
vection to coarse resolution global scales with parameterized convection (Pérez et al., 2011; Badia et al., 2017). MONARCH is
continuously developed at the Barcelona Supercomputing Center (BSC) with a focus on mineral dust and other aerosols (Pérez
et al., 2011; Haustein et al., 2012; Spada et al., 2013; Spada, 2015), atmospheric chemistry (Jorba et al., 2012; Badia and Jorba,
2015; Badia et al., 2017), emissions (HERMES, Guevara et al., 2019), data assimilation (Di Tomaso et al., 2017), workflow
management (Manubens-Gil et al., 2016), evaluation (Binietoglou et al., 2015; Ansmann et al., 2017), and operational forecast-
ing (Basart et al., 2019; Xian et al., 2019). Daily dust forecasts using MONARCH are produced at the BSC and made available
through the Barcelona Dust Forecast Center (a WMO Regional Specialized Meteorological Center with activity specialization
on Atmospheric Sand and Dust Forecast; https://dust.aemet.es/), the WMO SDS-WAS Regional Center for Northern Africa–
Middle East–Europe (NA-ME-E) (https://sds-was.aemet.es/) and the International Cooperative for Aerosol Prediction (ICAP)
(Sessions et al., 2015; Xian et al., 2019). Here we present recent developments on the representation of mineral dust processes
in MONARCH.

Mineral soil dust is the most abundant aerosol type in terms of global mass, competing only with sea salt (Textor et al.,
2006). Global dust emissions are estimated to range between 3300 and $9000\,\mathrm{Tg\,yr^{-1}}$ for particles smaller than $20\,\mu\mathrm{m}$ in
diameter (Kok et al., 2020). Soil dust is mainly emitted from arid and semi-arid regions, e.g. in Africa, the Middle East, central
and northeastern Asia, India, Australia, Patagonia and the southwestern United States, but can in principle be emitted from any
uncovered dry soil surface under windy conditions, e.g. from agricultural fields (Ginoux et al., 2012).

Mineral dust is emitted as soon as the forces that act to retain the soil particles at the surface (gravity and inter-particle
cohesion) are overcome either by atmospheric lifting forces generated by wind and turbulence (aerodynamic entrainment), or
by the force generated by other impacting particles, i.e. sand grains or particle aggregates (saltation bombardment/aggregate
disintegration) (Shao, 2008). Typically, soil particles in the silt and clay particle size range (diameter $< 63\,\mu\mathrm{m}$; Udden 1914;
Wentworth 1922) are considered dust, whereas larger particles are referred to as sand. Soil particles in the size-range 70–
$100\,\mu\mathrm{m}$ can typically be lifted most easily (Iversen and White, 1982; Shao and Lu, 2000). For larger diameters, the particle
weight is the predominant inhibitor. For smaller particles, inter-particle cohesion becomes more significant, but likely exhibits
stochastic behavior leaving a fraction of particles with substantially below-average cohesive forces (Shao and Klose, 2016).
Particles with diameters of around $70\,\mu\mathrm{m}$ and larger are mainly transported in ballistic trajectories along the surface (saltation).
The limit at which saltation is initiated, i.e. when the particle retarding forces are exceeded by the aerodynamic lifting forces,
is expressed as a threshold friction velocity, $u_{*t}$ $[\mathrm{ms^{-1}}]$. Saltation bombardment is typically most efficient at generating dust
emission (e.g. Shao et al., 1993; Houser and Nickling, 2001). Aerodynamic dust entrainment is typically less efficient because
of on-average higher cohesive forces for dust-sized particles compared to sand (or saltation) particles, but can be significant




under favorable atmospheric conditions and on long time scales, provided there is a sufficient supply of loose dust particles at the surface (e.g. Loosmore and Hunt, 2000; Macpherson et al., 2008; Chkhetiani et al., 2012; Klose and Shao, 2013; Li et al., 2014; Zhang et al., 2016).

Once airborne, mineral dust particles interact with short- and long-wave radiation through scattering and absorption (Boucher et al., 2013; Miller et al., 2014), which has important direct effects on the Earth's energy balance (Kok et al., 2018; Li et al., 55   2020). Dust particles are known to be efficient ice nuclei and can also act as cloud condensation nuclei (DeMott et al., 2003; Karydis et al., 2011; Cziczo et al., 2013; Kiselev et al., 2017). Nutrients transported with dust can create ecosystem responses due to, e.g. carbon uptake and storage (Jickells et al., 2005; Rizzolo et al., 2017; Kanakidou et al., 2018). Dust can cause respiratory and cardiovascular diseases (Meng and Lu, 2007), and can contribute to other ailments like meningitis (Pérez García-Pando et al., 2014) and valley fever (Tong et al., 2017). To study and quantify dust and its impacts, models that include 60   advanced dust representations, such as MONARCH, are key tools.

Existing dust emission parameterizations range from formulations that are strongly simplified (e.g. Ginoux et al., 2001) to those that aim to represent the physics of the emission processes (Shao, 2004; Klose et al., 2014; Kok et al., 2014b). The more simplified dust emission schemes are typically constrained by "preferential" source scaling functions and are commonly used in global, but also in regional models. Such constraints have significantly improved the skill of models by approximately 65   locating and enhancing dust emissions from prolific large-scale natural sources. However, these schemes are not very sensitive to changes in parameters known to affect dust emission (e.g. soil texture, soil moisture, land-surface properties), which at the same time can make models insensitive to changes in climate. In contrast, physics-based dust emission parameterizations are very sensitive to such changes, but need more detailed input. This detailed input has traditionally been difficult to observe and/or estimate, in particular globally, and errors in the description of, for example, surface properties, translate non-linearly into errors 70   in emitted and transported dust. How such errors compare with those arising when neglecting dust emission sensitivities entirely remains a subject of research and discussion. All in all, a clear benefit of physics-based schemes with detailed sensitivities is that input data sets can easily be updated as more data become available and hence future improvements are more likely, in particular for climate applications.

In this work, we introduce recent advancements in the treatment of mineral dust in MONARCH. The model now has diverse 75   available model configurations, in particular to estimate dust emission, which makes MONARCH unique among state-of-the-art models, and which makes it suitable for a variety of applications that range from process studies to operational forecasting and climate research. In the following sections, we briefly present the MONARCH modeling system and subsequently focus on the mineral dust cycle. We then demonstrate and evaluate MONARCH's dust modeling capabilities based on four annual global model runs.

## 80   2   The MONARCH model

MONARCH (previously known as NMMB/BSC-CTM) consists of advanced chemistry and aerosol packages coupled online with the Non-hydrostatic Multiscale Model on the B-grid (NMMB) (Janjic et al., 2001; Janjic and Gall, 2012), whose non-





**Table 1.** Available physics schemes in MONARCH.

| Process | Scheme | Reference |
|---|---|---|
| Microphysics | Ferrier (Eta) | Ferrier et al. (2002) |
| | Thompson | Thompson et al. (2008) |
| | WSM6 | Hong and Lim (2006) |
| Radiation | RRTMG | Iacono et al. (2008) |
| | GFDL | Fels and Schwarzkopf (1975) |
| Surface layer | NMMB similarity theory | Janjic (1994, 1996b) |
| Land surface | Unified NCEP/NCAR/AWFA NOAH | Ek et al. (2003) |
| | LISS | Vukovic et al. (2010) |
| Planteray Boundary Layer | Mellor–Yamada–Janjic | Janjic (1996a, 2002) |
| | GFS | Hong and Pan (1996) |
| Cumulus clouds | Betts-Miller-Janjic | Betts (1986); Betts and Miller (1986); Janjic (1994, 2000) |
| | Simplified Arakawa-Schubert | Han and Pan (2011) |

hydrostatic dynamical core allows running both global and regional simulations with embedded telescoping nests. The global model works on a latitude–longitude grid with polar filtering and the regional model on a rotated longitude–latitude grid. In both cases, the Arakawa B-grid and the hybrid pressure-sigma coordinate are used in the horizontal and vertical directions, respectively. The numerical schemes follow the principles described in Janjic (1977, 1979, 1984, 2003). The NMMB can be configured with a combination of different physics schemes (see Tab. 1). The configuration commonly used in production runs and in this work is as follows. Turbulence in the planetary boundary layer and the free troposphere is resolved using the Mellor–Yamada–Janjic (MYJ) level 2.5 turbulence closure scheme (Janjic, 2002). The surface layer scheme combines Monin–Obukhov similarity theory (Monin and Obukhov, 1954) with a viscous sublayer introduced over land and water (Zilitinkevich, 1965; Janjic, 1994, 1996b). The shortwave and longwave radiation fluxes are computed using the RRTMG radiation package (Iacono et al., 2008). The model includes the Ferrier scheme for grid-scale cloud microphysics (Ferrier et al., 2002), and the Betts–Miller–Janjic convective adjustment scheme (Betts, 1986; Betts and Miller, 1986; Janjic, 1994, 2000). The Unified NCEP/NCAR/AFWA NOAH (Ek et al., 2003) land surface model is used for the computation of heat and moisture surface fluxes.

The gas-phase chemistry in MONARCH solves the Carbon Bond 2005 chemical mechanism (CB05; Yarwood et al., 2005) extended with Toluene and Chlorine chemistry. The CB05 is well formulated for urban to remote tropospheric conditions and it considers 51 chemical species and solves 156 reactions. The photolysis rates are computed with the Fast-J scheme (Wild et al., 2000) considering the physics of each model layer (e.g., aerosols, clouds, absorbers such as ozone). The aerosol module in MONARCH describes the life cycle of dust, sea-salt, black carbon, organic matter (both primary and secondary), sulfate and nitrate aerosols (Spada, 2015). While a sectional approach is used for dust and sea-salt, a bulk description of the other aerosol



species is currently adopted (Spada, 2015). A simplified gas-aqueous-aerosol mechanism has been introduced in the module to account for the sulfur chemistry, the production of secondary nitrate - ammonium aerosol is solved using the thermodynamic equilibrium model EQSAM (Metzger et al., 2002), and a two-product scheme is used for the formation of secondary organic aerosols from gas-phase precursors. Different meteorology-driven emissions are computed on-line in MONARCH (i.e., mineral dust, sea salt and biogenic gas species). Sea salt emissions can be calculated with a wide range of available source functions (Spada et al., 2013) while the biogenic emissions are estimated with the MEGANv2.04 model (Guenther et al., 2006).

In addition to single model runs, MONARCH can be run in an ensemble mode for data assimilation applications, where the ensemble of model states is used to derive a flow-dependent background error covariance at the assimilation time, which evolves during the model forecast. The background error covariance is used to express model uncertainty within the data assimilation framework. Model uncertainty, together with observational uncertainty, is a key ingredient in the optimal integration of model simulations and observations for the production of an analysis that best represents the atmospheric state. The MONARCH ensemble is coupled with the local ensemble transform Kalman filter (LETKF) scheme (Hunt et al., 2007; Miyoshi and Yamane, 2007; Schutgens et al., 2010) for the estimation of dust analyses (Di Tomaso et al., 2017), as well as for the generation of dust reanalyses currently in production at the BSC (Di Tomaso et al., 2021, in prep.).

## 3 The mineral dust cycle in MONARCH

The dust module in MONARCH (previously known as NMMB/BSC-Dust), initially described by Pérez et al. (2011), solves the mass balance equation for dust taking into account the following processes: (1) dust generation and uplift by surface wind and turbulence (2) horizontal and vertical advection (3) horizontal diffusion and vertical transport by turbulence and convection (4) dry deposition and gravitational settling, and (5) wet removal by convective and stratiform clouds. The dust size distribution is represented with eight bins ranging up to $20\,\mu m$ in diameter: 0.2–0.36, 0.36–0.6, 0.6–1.2, 1.2–2.0, 2.0–3.6, 3.6–6.0, 6.0–12.0, and 12.0–20.0 $\mu m$. The effective and volume radii of each bin in the radiative and sedimentation schemes respectively (see Sec. 3.3, Tab. 6) are time-invariant and based on a lognormal distribution with mass median diameter of $2.524\,\mu m$ and geometric standard deviation of 2 (Schulz et al., 1998; Zender et al., 2003).

Our new developments presented below have mostly focused on aspect (1): dust generation and uplift by surface wind and turbulence. In particular, we have implemented and tested a variety of dust emission and drag partition parameterizations, along with new datasets for dust source areas, source type (i.e. natural and anthropogenic), surface roughness, and vegetation. Additional upgrades include the option to calculate dust extinction assuming non-spherical particle shape, as well as new diagnostic capabilities (output of 3-dimensional single-scattering albedo and extinction, clear-sky aerosol optical depth (clear-sky AOD) and AOD at satellite overpass times). In the following, we present the MONARCH dust module. We first describe the treatment of dust emission, summarizing previous and detailing new developments. Then, we recapitulate the implementation of dust transport and deposition, and interactions with radiation.





**Table 2.** Summary of the seven available dust emission schemes in MONARCH.

| Dust emission scheme/Reference | Abbreviation | Approach |
|---|---|---|
| Marticorena and Bergametti (1995) | MB95 | Dust emission based on saltation flux and soil texture |
| Ginoux et al. (2001, modified) | G01-U/G01-UST | Dust emission based on a topographic dust source function |
| Shao (2001) | S01 | Dust emission based on volume removal by saltation |
| Shao (2004) | S04 | Dust emission based on volume removal by saltation (parameterized saltation bombardment efficiency) |
| Shao et al. (2011) | S11 | Dust emission based on volume removal by saltation (reduced form) |
| Kok et al. (2014b) | K14 | Dust emission based on brittle fragmentation by saltation |
| Klose et al. (2014) | KS14 | Dust emission by aerodynamic entrainment |

### 3.1 Dust emission and lower boundary conditions

Several different parameterizations of dust emission are available in MONARCH, which cover different paradigms and range
from more simplified to more physics-based descriptions. To describe dust emission generated by saltation, MONARCH in-
cludes the parameterizations from Marticorena and Bergametti (1995) (MB95), Ginoux et al. (2001) with modifications detailed
below (G01), Shao (2001, 2004) (S01, S04), Shao et al. (2011, Eq. 34) (S11), and Kok et al. (2014b) (K14). To describe dust
emission in the absence of saltation, MONARCH includes the aerodynamic dust entrainment scheme developed by Klose et al.
(2014) (KS14). The seven available dust emission schemes in MONARCH are summarized in Tab. 2.

### 3.1.1 Dust emission flux

In saltation-based dust emission schemes, the vertical dust emission flux $F$ depends on the horizontal flux of saltating soil
particles or particle aggregates. In the MB95 scheme, $F$ is directly proportional to the total streamwise saltation flux $Q$,

$$F_{\mathrm{MB95}} = S\alpha_q Q. \tag{1}$$

In MONARCH, the vertical-to-horizontal-flux ratio $\alpha_q$ can either depend on the clay content of the parent soil as originally
proposed in Marticorena and Bergametti (1995), or on the soil texture as proposed in Tegen et al. (2002) and described in
Pérez et al. (2011). In the latter case, which is the default in MONARCH, $\alpha_q$ is determined as a mass-weighted average of the
vertical-to-horizontal-flux ratios of four soil particle size classes with mean diameters 2, 15, 160, and 710 µm. $S$ is a globally
variable dust source scaling function, which was not part of the original formulation in Marticorena and Bergametti (1995), but
which is introduced here as it was found to lead to improved results (Pérez et al., 2011, see also Sec. 3.1.6). The dust emission
flux resulting from Eq. (1) is a bulk flux, which we distribute across particle sizes using a predefined particle size-distribution
(Sec. 3.1.5).

The G01 dust emission scheme does not include an explicit formulation of $Q$. It seeks to avoid the need for detailed descrip-
tions of soil characteristics and instead introduces a topography-based dust source function, $S$, representing the availability of





sediment. The dust emission flux is originally obtained as

$$F_{\text{G01-U}}(d_i) = \begin{cases} s_{\text{bare}}C_{\text{G01}}Ss_p u_{10\text{m}}^2(u_{10\text{m}} - u_t) & u_{10\text{m}} > u_t \\ 0 & \text{otherwise} \end{cases}, \tag{2}$$

where $C_{\text{G01}}$ is a dimensional factor, $s_{\text{bare}}$ is the bare soil fraction (see Sec. 3.1.4), $u_{10\text{m}}$ is the $10\,\text{m}$ wind speed, and $u_t$ is a threshold wind speed below which no dust emission occurs (Sec. 3.1.3). Note that $s_{\text{bare}}$ was included in $S$ in the original formulation. We apply $s_{\text{bare}}$ to all schemes in MONARCH. The bulk dust emission flux $F$ is distributed across particle size classes using predefined fractions $s_p$ (Sec. 3.1.5). To ease comparison with other schemes, we also implemented a modified version of the G01 scheme, which estimates $F$ using friction velocity and threshold friction velocity, $u_*$ and $u_{*t}$ (G01-UST), instead of $u_{10\text{m}}$ and $u_t$ (G01-U), such that

$$F_{\text{G01-UST}}(d_i) = \begin{cases} s_{\text{bare}}C_{\text{G01}}Ss_p u_*^2(u_* - u_{*t}) & u_* > u_{*t} \\ 0 & \text{otherwise} \end{cases}. \tag{3}$$

In both implementations, G01-U and G01-UST, we introduced additional modifications on, respectively, $u_t$ and $u_{*t}$, and on $s_p$ as described in Secs. 3.1.3 and 3.1.5.

The S01 scheme is a physics-based dust emission scheme, which calculates size-resolved dust emission based on the soil volume removed by impacting saltation particles and explicitly considers aggregate disintegration as a dust emission process in addition to saltation bombardment. The emission of dust particles of size $d_i$ by saltation particles of size $d_s$ is given by (Shao, 2001, Eq. 52)

$$\tilde{F}_{\text{S01}}(d_i, d_s) = s_{\text{bare}}c_y\left[(1-\gamma) + \gamma\sigma_{p_i}\right]\frac{gQ_s}{u_*^2 m_{p_s}}\left(\rho_b\eta_{fi}\Omega + \eta_{ci}m_{p_s}\right), \text{ with} \tag{4}$$

$$\gamma = \exp\left[-\kappa\left(u_* - u_{*t}\right)^3\right], \tag{5}$$

where $c_y$ and $\kappa$ are coefficients, $\eta_{fi}$, $\eta_{ci}$, and $\eta_{mi}$ are the total, aggregated, and free dust fractions at diameter $d_i$, $\sigma_{p_i} = \eta_{mi}/\eta_{fi}$, $Q_s$ the saltation flux of particles with diameter $d_s$, $m_{p_s} = m_p(d_s)$ the mass of a spherical particle with diameter $d_s$ assuming a density of $\rho_{ps} = 2650\,\text{kg m}^{-3}$, $\rho_b \approx 1000\,\text{kg m}^{-3}$ the soil bulk density, $g$ gravitational acceleration, and $\Omega$ the soil volume removed by a saltating particle of size $d_s$ (Lu and Shao, 1999, Eq. 8). The removed soil mass is given by $m_\Omega = \rho_b\Omega$. The bare soil fraction $s_{\text{bare}}$ was added here for implementation in MONARCH. The dust fractions $\eta_{fi}$, $\eta_{ci}$, and $\eta_{mi}$ can be estimated from the minimally and fully dispersed particle-size distributions (PSDs), $p_m(d_i)$ and $p_f(d_i)$, as $\eta_{mi} = \int_{d_{i0}}^{d_{i1}} p_m(d_i)\delta d_i$ and $\eta_{fi} = \int_{d_{i0}}^{d_{i1}} p_f(d_i)\delta d_i$, where $d_{i0}$ and $d_{i1}$ are the lower and upper limits of the particle-size bin corresponding to $d_i$. The aggregated dust fraction follows as $\eta_{ci} = \eta_{fi} - \eta_{mi}$. The $\gamma$-function (Eq. 5) and therein the parameter $\kappa$ determine how easily a soil is disaggregated (Shao et al., 2011; Klose et al., 2019, see also Sec. 3.1.2). Here, we use $\kappa = 1$ globally. A spatially variable, for example soil-texture dependent, $\kappa$ could be easily implemented, if future investigations support such a dependency. The emission flux of dust particles with diameter $d_i$, i.e. for all saltation particle sizes, is obtained as

$$F(d_i) = \sum_{ds=d_{\text{max}}}^{\infty} \tilde{F}(d_i, d_s), \tag{6}$$





with $d_{\mathrm{max}} = 20\,\mu\mathrm{m}$ in MONARCH.

The S04 scheme is a simplification of the S01 scheme in which the saltation bombardment efficiency, $\sigma_m = m_\Omega/m_{p_s}$, is approximated as (Shao, 2004, Eq. 11)

$$\sigma_m = 12u_*^2\frac{\rho_b}{P}\left(1 + 14u_*\sqrt{\frac{\rho_b}{P}}\right), \tag{7}$$

with soil plastic pressure $P$. The larger $u_*$, the more soil mass is ejected by a saltation particle impact for a given soil. The dust emission flux is given by (Shao, 2004, Eq. 6)

$$\tilde{F}_{\mathrm{S04}}\left(d_i, d_s\right) = s_{\mathrm{bare}}c_y\eta_{fi}\left[(1-\gamma) + \gamma\sigma_{p_i}\right](1 + \sigma_m)\frac{gQ_s}{u_*^2}, \tag{8}$$

and $F(d_i)$ follows from Eq. 4 assuming $\eta_{fi} \approx \eta_{ci}$. Based on a detailed comparison with field measurements, a basic version of the scheme (denoted here as S11) was suggested by Shao et al. (2011, Eq. 34), which makes use of the total (instead of size-resolved) saltation flux:

$$F_{\mathrm{S11}}\left(d_i\right) = s_{\mathrm{bare}}c_y\eta_{mi}\left(1 + \sigma_m\right)\frac{gQ}{u_*^2}. \tag{9}$$

Shao et al. note, however, that this simplification may be specific to the experimental data set, which had a narrow soil PSD.

The K14 dust emission scheme uses the concept of the fragmentation of brittle material. It is also a physics-based dust emission scheme that includes a dynamical dependency of soil erodibility on threshold friction velocity. Although the kinetic energy supplied by saltating particles is taken into account in the scheme, it does not include $Q$ explicitly. The K14 dust emission flux is given as

$$F_{\mathrm{K14}}\left(d_i\right) = C_e s_{\mathrm{bare}} f_{\mathrm{clay}}\frac{\rho_a\left(u_*^2 - u_{*t}^2\right)}{u_{*st}}\left(\frac{u_*}{u_{*t}}\right)^{C_\alpha\psi_*} \qquad u_* > u_{*t} \tag{10}$$

$$\psi_* = \frac{u_{*st} - u_{*st0}}{u_{*st0}} \tag{11}$$

where $f_{\mathrm{clay}}$ is the clay fraction (from STATSGO-FAO inventory, see Sec. 3.1.6), $\rho_a$ is air density, $u_{*st} = u_{*t}\sqrt{\rho_a/\rho_{a0}}$ with $\rho_{a0} = 1.225\,\mathrm{kg\,m^{-3}}$ is a standardized threshold friction velocity, $C_\alpha$ is a constant coefficient, and $C_e$ is a $u_{*st}$-dependent coefficient representing soil erodibility.

The previously described dust emission schemes all describe dust emission related to saltation (through saltation bombardment and/or aggregate disintegration), whereas the parameterization from KS14 describes dust emission by aerodynamic forces, i.e. without saltation as an intermediate process. Direct aerodynamic dust entrainment is expected to be most relevant under (convective) turbulent atmospheric conditions, when average wind speeds are small and saltation is absent or only sporadic (Klose and Shao, 2013). The KS14 scheme is a stochastic parameterization, which represents both the aerodynamic lifting forces due to instantaneous momentum fluxes and the interparticle cohesive forces as probability density functions (pdfs). For a given lifting force $f$ and cohesive force $f_i$, the dust emission flux of particles with diameter $d_i$ is given as (Klose et al., 2014, Eq. 1)

$$\tilde{F}\left(d_i\right) = \begin{cases} \frac{\alpha_N}{2D_v}\left[-w_t m_{p_i} + T_p\left(f - f_i\frac{d_i}{D_v}\right)\right] & f > f_t \\ 0 & \text{otherwise} \end{cases}, \tag{12}$$





where $D_v$ is the depth of the viscous sublayer, $w_t$ the particle terminal velocity, $T_p$ the particle response time, and $\alpha_N = N_p D_v$ is a parameter of dimension $[\mathrm{m}^{-2}]$ representing the particle number concentration integrated over $D_v$, $N_p$. $\alpha_N$ is a function of

particle size. Dust is emitted only if the lifting force $f$ exceeds the retarding force $f_t = f_i + m_p g$. The dust emission flux for a given particle size $d_i$ follows through integration over the pdfs of the cohesive forces (depending on $d_i$), $p_i(f_i)$, and of the lifting force, $p(f)$. Taking into account the bare soil fraction,

$$F(d_i) = s_{\text{bare}} \int_0^\infty \left[ \int_0^f \tilde{F}(d_i) \cdot p_i(f_i)\,\mathrm{d}f_i \right] p(f)\,\mathrm{d}f. \tag{13}$$

Finally, the total dust emission flux is obtained through integration over the area particle-size distribution, $p_A(d_i)$:

$$F = \int_0^{d_{\max}} F(d_i)\, p_A(d_i)\, \delta d_i, \tag{14}$$

where $d_{\max}$ is the maximum particle size considered (20 µm in MONARCH) and $\delta$ indicates the differential. $p_A$ is inferred from the minimally dispersed PSD, $p_m$, based on Eq. (5) in Klose et al. (2014).

### 3.1.2   Saltation flux

For the schemes that contain explicit representations of the saltation flux (MB95, S01, S04, S11), the saltation flux of particles

with diameter $d_s$, $Q_s$, is calculated following Kawamura (1964) (same as White, 1979) as

$$Q_s(d_s) = c_Q \frac{\rho_a}{g} \left( 1 + \frac{u_{*t}(d_s)}{u_*} \right) \left( 1 - \frac{u_{*t}(d_s)^2}{u_*^2} \right) \qquad \text{for } u_* > u_{*t}(d_s), \tag{15}$$

where $c_Q$ is a coefficient, $u_{*t}(d_s)$ the threshold friction velocity for particles with diameter $d_s$, and $u_*$ the friction velocity for the bare surface. In Eq. (15), the saltation of particles of different sizes is treated independently. For a soil that consists of a mixture of different sized loose particles of sufficient availability, particle impacts can cause saltation in a wider size range

than it would be expected based on $u_{*t}(d_s)$ (Ungar and Haff, 1987; Martin and Kok, 2019). In the MB95 implementation, the total saltation flux $Q$ is used and obtained as a weighted average taking into account the relative surface area of particles in four size classes (see Sec 3.1.1 for mean diameters) as a function of soil texture (Pérez et al., 2011, Eq. 2). The S11 scheme is also based on the total saltation flux, but takes a different approach and obtains $Q$ by weighting $Q_s$ with the particle-size distribution estimated for airborne sediment, $p_s(d_s)$, as

$$Q = \int Q_s(d)\, p_s(d)\, \delta d, \text{ with} \tag{16}$$

$$p_s(d_s) = \gamma p_m(d_s) + (1 - \gamma) p_f(d_s). \tag{17}$$

The $\gamma$-function (Eq. 5) determines how rapidly $p_s$ approaches $p_f$ with increasing $u_*$, i.e. how easily soil aggregates are disintegrated (Shao et al., 2011; Klose et al., 2019). Both, $p_m$ and $p_f$, are estimated for each soil texture class as a combination of up to four lognormal distributions. The coefficients used for those distributions are given in Tab. 3. PSDs are calculated with



**Table 3.** Coefficients for minimally-dispersed particle-size distributions as assigned to the 12 USGS soil texture classes. Each PSD is composed of four lognormal distributions $p_1$, $p_2$, $p_3$, and $p_4$. Coefficients are taken from Klose (2014), Table 3, unless otherwise indicated.

| | $p_1$ | | | $p_2$ | | | $p_3$ | | | $p_4$ | | |
| | $w$ | $\overline{\ln d}$ | $\sigma$ | $w$ | $\overline{\ln d}$ | $\sigma$ | $w$ | $\overline{\ln d}$ | $\sigma$ | $w$ | $\overline{\ln d}$ | $\sigma$ |
|---|---|---|---|---|---|---|---|---|---|---|---|---|
| sand[1] | 0.50 | 5.50 | 0.43 | 0.42 | 6.07 | 0.42 | 0.07 | 4.22 | 0.60 | 0.01 | 2.03 | 0.38 |
| loamy sand[2] | 0.66 | 5.56 | 0.44 | 0.26 | 6.03 | 0.31 | 0.07 | 6.43 | 0.21 | 0.01 | 3.82 | 0.33 |
| sandy loam | 0.60 | 6.07 | 0.41 | 0.32 | 5.18 | 0.75 | 0.05 | 6.07 | 0.12 | 0.02 | 6.66 | 0.10 |
| silt loam[3] | 0.48 | 5.44 | 0.37 | 0.42 | 4.57 | 0.75 | 0.08 | 6.22 | 0.14 | 0.02 | 3.99 | 0.17 |
| silt | 0.50 | 4.33 | 0.45 | 0.31 | 3.58 | 1.07 | 0.17 | 4.14 | 0.19 | 0.03 | 5.21 | 0.19 |
| loam[3] | 0.46 | 6.08 | 0.32 | 0.35 | 5.55 | 0.71 | 0.11 | 4.34 | 0.95 | 0.08 | 4.36 | 0.24 |
| sandy clay loam[3] | 0.71 | 5.23 | 0.53 | 0.20 | 4.30 | 0.27 | 0.06 | 6.17 | 0.27 | 0.03 | 3.51 | 0.37 |
| silty clay | 1.26 | 4.80 | 0.38 | 0.81 | 5.25 | 0.30 | 0.45 | 5.12 | 1.26 | 0. | 0. | 0. |
| clay loam[4] | 0.50 | 5.17 | 0.31 | 0.25 | 4.62 | 0.28 | 0.24 | 5.02 | 0.93 | 0.01 | 4.91 | 0.10 |
| sandy clay | 1.03 | 4.31 | 0.43 | 0.96 | 3.95 | 1.78 | 0.31 | 4.14 | 0.17 | 0. | 0. | 0. |
| silty clay | 0.53 | 4.53 | 0.49 | 0.27 | 4.92 | 0.20 | 0.14 | 3.90 | 0.81 | 0.06 | 4.58 | 0.16 |
| clay | 0.67 | 5.31 | 0.39 | 0.24 | 4.59 | 0.63 | 0.06 | 3.31 | 1.17 | 0.03 | 5.39 | 0.10 |

[1]Coefficients for samples from Site D in Klose et al. (2019) (PSD$_{LEM}$)

[2]Coefficients from Table 3 of Klose et al. (2019) (PSD$_{LEM}$).

[3]Different sample used as reference than in Table 6.1 of Klose (2014), but same underlying data set.

[4]Sandy clay loam in Table 6.1 of Klose (2014)

60 size-bins distributed logarithmically using a quarter-$\varphi$ scale (Krumbein, 1934, 1938) with reference diameter $2000\,\mu m$. The S01 and S04 schemes directly use the spectral, i.e. size-resolved, saltation flux from Eq. (15) (cf. Sec. 3.1.1). The G01 and K14 schemes do not contain explicit formulations for saltation flux.

### 3.1.3 Threshold friction velocity and soil moisture correction

The implementation of the threshold friction velocity for ideal (dry) conditions, $u_{*t0}(d_i)$, varies depending on the dust emission scheme and its requirements. In the MB95 implementation, we use the relationship from Iversen and White (1982) for the four saltation size classes (cf. Sec. 3.1.2) as described in Pérez et al. (2011). The original parameterization of Ginoux et al. (2001) estimates dust emission based on $10\,m$ wind speed instead of friction velocity (Sec. 3.1.1) and specifies a threshold wind speed for each dust size bin, which can typically be expected to be larger than for saltation-particle sizes (see Sec. 1). In combination with the relatively simple and constant distribution of soil particles across clay and silt particle-sizes (Ginoux et al., 2001), this dust-size dependent threshold wind speed leads to a more variable particle-size distribution at emission. Here, we revise this implementation and specify the entrainment threshold for saltation in G01-UST as

$$u_{*t_{d0}} = \min_{d_i} [u_{*t0}(d_i)] \tag{18}$$





based on the theoretical expression for $u_{*t0}(d_i)$ from Shao and Lu (2000). In the G01-U implementation, we use a fixed minimal threshold of $u_{t_{d0}} = 5\,\mathrm{m\,s^{-1}}$. To obtain a more realistic PSD at emission in combination with a dust-particle size independent

entrainment threshold, we replace the PSD described in Ginoux et al. (2001) with that of Kok (2011a) (see Sec. 3.1.5). The K14 scheme also makes use of a particle-size independent threshold friction velocity and in MONARCH, $u_{*t_{d0}}$ for K14 is obtained based on Iversen and White (1982) for $70\,\mu\mathrm{m}$, a diameter in the size-range where $u_{*t0}$ becomes minimal. In the implementations of the S01, S04, and S11 dust emission schemes, $u_{*t0}(d_i)$ is described as in Shao and Lu (2000) and the minimum value (Eq. 18) is used in Eq. (5).

Models are known to underestimate the tail of the wind speed distribution by different degrees depending on their resolution. This is particularly relevant for dust emission (Cakmur et al., 2004; Cowie et al., 2015). If the frequency of occurrence of wind speeds or friction velocities above the threshold for particle entrainment is underestimated, dust emission will be underestimated, too. For coarse model resolutions (temporal or spatial), this underestimation might be considerable in some regions, for example in areas with frequent moist convection or pronounced topography. In some models, this effect is mitigated by

introducing sub-grid scale wind variability (e.g. Cakmur et al., 2004; Lunt and Valdes, 2002). In our model, we included an optional constant scaling parameter, $c_{\mathrm{thr}} \leq 1$, such that the final threshold friction velocity for dry conditions, $u_{*t_{\mathrm{dry}}}$ is

$$u_{*t_{\mathrm{dry}}}(d_i) = c_{\mathrm{thr}} \cdot u_{*t0}(d_i). \tag{19}$$

As a result, dust emission is initiated more often and over larger areas.

When the soil is moist, the threshold friction velocity above which particles are lifted is higher than under dry conditions,

because soil-water capillary forces increase the cohesion between the soil particles (Chepil, 1956; Zimon, 1982; Chen et al., 1996). This is implemented by first estimating the threshold friction velocity for dry conditions, $u_{*t_{\mathrm{dry}}}$, and then applying a correction factor, $f_w > 1$, to obtain the threshold friction velocity for the given (moist) conditions (McKenna Neuman and Nickling, 1989; Fécan et al., 1999; McKenna Neuman, 2003; Cornelis et al., 2004a, b; Klose et al., 2014):

$$u_{*t}(d_i) = u_{*t_{\mathrm{dry}}}(d_i) \cdot f_w. \tag{20}$$

In MONARCH, the soil moisture corrections from Belly (1964), Fécan et al. (1999), and Shao and Jung (unpublished manuscript, 2000; see Klose et al. 2014) are available in combination with all saltation-based schemes. In the aerodynamic entrainment scheme (KS14), a soil moisture correction for inter-particle cohesive force rather than threshold friction velocity is used (Klose et al., 2014). The options to account for the impact of soil moisture on dust emission in MONARCH are summarized in Tab. 4 and further detailed below.

The soil moisture correction from Belly (1964) is implemented as described in Ginoux et al. (2001):

$$f_{w_B} = \begin{cases} 1.2 + 0.2\log_{10}\left(\max\left(0.001, c_{f_1} \cdot \theta\right)\right) & \theta < 0.5 \\ f_{w_{B\mathrm{wet}}} & \text{otherwise} \end{cases}, \tag{21}$$

where $\theta$ is the volumetric soil moisture [$\mathrm{m^3\,m^{-3}}$], $f_{w_{B\mathrm{wet}}}$ is a large value prohibiting particle movement (here $f_{w_{B\mathrm{wet}}} = 100$), and $c_{f_1}$ is an optional calibration factor described below. This correction is used as the default for the G01 scheme. The soil





**Table 4.** Summary of available options in MONARCH to account for soil moisture in the particle entrainment threshold.

| Soil moisture correction reference | Description | Remark |
|---|---|---|
| Belly (1964) | as in Ginoux et al. (2001) | Default for G01 |
| Fécan et al. (1999) | static coefficients; gravimetric soil moisture after Zender et al. (2003); sand fraction from Tegen et al. (2002) | Default for MB95 |
| Fécan et al. (1999) | static coefficients; gravimetric soil moisture after Zender et al. (2003); sand fraction from Kok et al. (2014b) | Default for K14 |
| Shao and Jung (2000, unpubl.)/Klose et al. (2014) | soil-texture dependent coefficients; volumetric soil moisture | Default for S01, S04, S11 |
| Klose et al. (2014) | soil-texture dependent coefficients; volumetric soil moisture; applied to cohesive force | Default for KS14 |

moisture correction after Fécan et al. (1999) is implemented as

$$f_{w_{F_w}} = \sqrt{1 + a\left(c_{f_1} \cdot w - c_{f_2} \cdot w_r\right)^b} \quad c_{f_1}w > c_{f_2}w_r, \tag{22}$$

with gravimetric soil moisture content $w$ [%], gravimetric air-dry residual soil moisture content $w_r$ [%], and coefficients $a = 1.21$ and $b = 0.68$ (Fécan et al., 1999). $w_r$ is obtained based on Eq. (14) in Fécan et al. (1999). The conversion from volumetric soil moisture content $\theta$ to $w$ is implemented as described by Zender et al. (2003, Eqs. 7-9):

$$w = 100 \cdot \theta \frac{\rho_l}{\rho_{bd}}, \tag{23}$$

$$\rho_{bd} = \rho_{pa}\left(1 - \theta_{s_z}\right), \text{ and} \tag{24}$$

$$\theta_{s_z} = 0.489 - 0.126 M_{\text{sand}}. \tag{25}$$

Here, $\rho_l$ is the density of water, $\rho_{bd}$ is the bulk density of dry soil, $\rho_{pa}$ is the average soil particle density (here $\rho_{pa} = 2500\,\text{kg m}^{-3}$), $\theta_{s_z}$ is the volumetric soil moisture at saturation, and $M_{\text{sand}}$ is the sand fraction in the soil (Pérez et al., 2011, Tab. 1). The factor 100 converts soil moisture content from $\text{kg kg}^{-1}$ into %. As the top-layer soil moisture in models is usually

obtained for a layer of several centimeters and is therefore typically higher than at the actual surface–atmosphere interface (which is relevant for dust emission), the soil moisture correction $f_w$ using the model's soil moisture is often too high and precludes dust emission. An optional calibration factor, $c_{f_1}$ or $c_{f_2}$, can therefore be applied if needed. The coefficient $c_{f_1} \leq 1$ directly reduces the soil moisture in Eq. (22) (e.g. Shao et al., 2010). Soil moisture remains unmodified outside of Eq. (22). Alternatively, the coefficient $c_{f_2} \geq 1$ (Zender et al., 2003) instead increases the air-dry soil moisture. Both coefficients have the

effect to reduce $f_w$. We recommend using either $c_{f_1}$ or $c_{f_2}$, but not both at the same time.

Shao and Jung (2000, unpublished manuscript) and Klose et al. (2014) developed a soil moisture correction similar to that of Fécan et al. (1999), but based on the soil-water retention curve from Brooks and Corey (1964) rather than that from Gardner





(1970). Including the optional coefficient $c_{f_1}$, the correction is

$$f_{w_K} = \sqrt{1 + \frac{h_w}{\psi_s} \left( \frac{c_{f_1} \theta - \theta_r}{\theta_s - \theta_r} \right)^{\beta}} \qquad c_{f_1} \theta > \theta_r, \tag{26}$$

where $\theta_r$ is the volumetric air-dry residual soil moisture, $h_w$ is a function combining different constants, and $\psi_s$ is the saturation

capillary pressure head (Klose et al., 2014). Eq. (26) is consistent with Eq. (22), as can be seen when setting

$$\alpha = \frac{h_w}{\psi_s} (\theta_s - \theta_r)^{-\beta}. \tag{27}$$

Note that the volumetric soil moisture $\theta$ [m³ m⁻³] is used in Eq. (26). $\theta_s$ is the saturation (volumetric) soil moisture. The

values for $\alpha$, $\beta$, and $\theta_r$ were obtained in Shao and Jung (2000) through fitting with observations and were published in Klose

et al. (2014, Tab. 1). The optional tuning constant $c_{f_2}$ was not implemented in Eq. (26) for simplicity as this would require

modifying $\alpha$.

The KS14 scheme does not include a deterministic threshold friction velocity for entrainment. Instead, the particle retarding

forces that need to be overcome for particle lifting are composed of the gravitational force and the interparticle cohesive force

which is assumed to follow a probabilistic distribution for a given particle size (see Sec. 3.1.1). Considering the lower limit of

this probabilistic representation as the threshold for free dust entrainment would lead to a significantly lower threshold friction

velocity in the dust-size range compared with the average deterministic behavior described above (Shao and Klose, 2016).

Soil moisture directly affects the cohesive force in the KS14 scheme. While under dry conditions, the variance of the cohesive

force is assumed to be relatively large due to variations in the particles' properties, capillary forces become dominant with

increasing soil wetness, i.e. the mean cohesive force increases and the variance decreases (Klose et al., 2014). The formulation

for capillary cohesive force, $f_{i_c}$, developed in Klose et al. (2014) is given by

$$f_{i_c} = \alpha \cdot m_p g \frac{\sin \xi}{\sin 2\xi} (\theta - \theta_r)^{\beta}, \tag{28}$$

where $m_p$ is the particle mass and $\xi$ the resting angle (here $\xi = 45°$). Equation (28) is based on Brooks and Corey (1964),

McKenna Neuman and Nickling (1989), and McKenna Neuman (2003).

### 3.1.4  Surface roughness, drag partition, and cover

Surface roughness through, e.g., vegetation, pebbles or rocks, absorbs momentum from the air flow and reduces the atmospheric

momentum available for particle entrainment. We account for this drag partitioning using either the scheme of Raupach et al.

(1993) or that of Marticorena and Bergametti (1995) with a correction published in King et al. (2005) (Tab. 5). Typically the

drag partition correction is applied to $u_{*t}$, which is phenomenologically, but not physically, correct as discussed in Kok et al.

(2014b). For use with all schemes in MONARCH, we apply the drag partition correction, $f_v < 1$, on the friction velocity

$u_{*NMMB}$ provided by the atmospheric model NMMB, such that the friction velocity acting on the erodible surface and used in

Eq. (15) is

$$u_* = f_v \cdot u_{*NMMB}. \tag{29}$$





**Table 5.** Summary of available options in MONARCH to account for surface roughness in particle entrainment.

| Roughness correction/Reference | Description of input data |
| --- | --- |
| Marticorena and Bergametti (1995); King et al. (2005) | static roughness length (Prigent et al., 2012) and dynamic roughness length from monthly MODIS LAI (Myeni et al., 2015) |
| Raupach et al. (1993) | dynamic frontal area index from monthly vegetation cover (Guerschman et al., 2015, photosynthetic and non-photosynthetic vegetation) or AVHRR (Gutman and Ignatov, 2010, green vegetation) |

In the parameterization of Raupach et al. (1993), the ratio $f_v$ between the friction velocity acting on the erodible surface and the total friction velocity supplied by the atmosphere is given as

$$f_{v_R} = \left(\frac{\tau_s''}{\tau}\right)^{1/2} = \left[\frac{1}{(1 - m\sigma_v\lambda)(1 + m\beta_R\lambda)}\right]^{1/2}, \tag{30}$$


where $\tau$ is the total stress, $\tau_s'' = \tau_s'(m\lambda)$ is the maximum surface stress on the exposed area estimated from the average surface shear stress on the exposed area, $\tau_s'$, for a surface with lower roughness density using the constant $m \leq 1$, $\sigma_v$ is the ratio of roughness-element basal to frontal area, and $\beta_R$ is the ratio of roughness-element to surface drag coefficients. Here we chose $\sigma_v = 1$, $\beta_R = 200$, and $m = 0.5$ (Shao et al., 2015). We estimate the frontal area index, $\lambda$, based on the vegetation cover fraction as (Shao et al., 1996)


$$\lambda = -c_\lambda \ln(1 - \eta), \tag{31}$$

where $c_\lambda$ is a coefficient. If the roughness elements are uniformly distributed and isotropically oriented, $c_\lambda = 1$ (Raupach et al., 1993; Shao et al., 1996). As this is typically not the case, a value of $c_\lambda = 0.35$ was proposed by Shao et al. (1996) based on measurements for stubble roughness. Stubble roughness can typically be associated with agricultural land use for which vegetation and its remains after the growing season are still relatively homogeneously distributed. An even smaller value for $c_\lambda$, which leads to a weaker effect of vegetation cover in the drag partition correction, may be more appropriate for roughness elements that are distributed heterogeneously, as it is typical in semi-arid regions. Here we choose $c_\lambda = 0.2$. In MONARCH, $\lambda$ can be estimated using Eq. (31) based on monthly satellite-based retrievals of photosynthetic and non-photosynthetic vegetation cover (PV and NPV) (Guerschman et al., 2009, 2015), interpolated to the day of simulation (used as $\eta$ in Eq. 31). Although NPV is intended to represent only vegetation components, it may also include some geological features, which is advantageous for our purposes. Monthly climatologies of the same data set (2003–2017) and also of green vegetation cover estimated from AVHRR (1985–1990, Gutman and Ignatov, 2010) are also available.



Figure 1 (a–c) shows annual averages for 2012 of PV, NPV and $f_{v_R}$. With the parameter settings as described above, only areas in northern Africa, the Middle East, and western East Asia (Taklamakan desert) experience a low or moderate roughness correction. Areas in other parts of East Asia, Central Asia, Australia, as well as parts of North and South America and southern Africa show a stronger correction, but one which can still allow dust emission under strong wind conditions. Dust emission




**Figure 1.** Comparison of roughness input and drag partition approaches: (a) and (b) show, respectively, annual averages for 2012 of photosynthetic and non-photosynthetic vegetation cover fractions (Guerschman et al., 2015), which we use to obtain the roughness correction $f_{v_R}$ (label DPR) based on Raupach et al. (1993) (parameters $c_\lambda = 0.2$, $\beta = 200$, $m = 0.5$, $\sigma = 1$) shown in (c); (d) displays the annual average MODIS leaf-area index and (e) static aerodynamic roughness length (Prigent et al., 2012), which we utilize for the roughness correction $f_{v_M}$ (label DPM) after Marticorena and Bergametti (1995) (parameters $z_{0s} = (2 \times 650 \cdot 10^{-4}/30)$ cm, $X = 12,255$ (MacKinnon et al., 2004)).

from other areas is typically suppressed by a larger vegetation coverage using this drag partition parameterization and the given parameters. Variations in the parameters used for $f_{v_R}$ will lead to changes in the roughness correction, particularly in areas with moderate vegetation coverage.





In the formulation from Marticorena and Bergametti (1995), $f_v$ is given by

$$f_{v_M} = 1 - \frac{\ln(z_0/z_{0s})}{\ln(0.7(X/z_{0s}))}, \tag{32}$$

where $z_0$ is the aerodynamic roughness length, $z_{0s}$ is the smooth aerodynamic roughness length, and $X$ is a parameter related to the distance downwind from an individual obstacle. As the surface becomes rougher (corresponding to increasing $z_0$), $f_v$ becomes smaller and the stress on the erodible surface decreases, reducing emission. The smooth roughness length $z_{0s}$ is

estimated as

$$z_{0s} = \frac{2d_c}{30}, \tag{33}$$

where $d_c = 650 \cdot 10^{-4}$ cm is assumed to be the coarsest diameter of particles in the soil bed (Sherman, 1992; Pierre et al., 2014). The aerodynamic roughness length $z_0$ is obtained globally in MONARCH as a combination of two different data sets. In arid regions, we use a static roughness, $z_{0\text{stat}}$, which is derived from satellite microwave backscatter (ASCAT) and visible/near-

infrared reflectances (PARASOL) (Prigent et al., 2012). In semi-arid regions, including natural vegetation and cultivated areas, we estimate a time-varying or "dynamic" roughness ($z_{0\text{dyn}}$) based on the dimensions of green vegetation characterized using the MODIS Leaf Area Index (LAI). The calculation of $z_{0\text{dyn}}$ is based on empirical relationships from Marticorena et al. (2006):

$$z_{0\text{dyn}} = \begin{cases} h \cdot 10^{1.3\log\lambda + 0.66} & \lambda < 0.041 \\ h \cdot 10^{-1.16} & \lambda \geq 0.041 \end{cases}, \tag{34}$$

where $h$ is the vegetation height and $\lambda$ the roughness density (or frontal area index), defined as $\lambda = n \cdot a_f$, where $n$ is the

number density of roughness elements (number per unit area) having a frontal area $a_f$. $\lambda$ is calculated assuming patches of vegetation of diameter $D_\eta = 5\,\text{m}$, the number of which increases with the vegetation cover fraction $\eta$, $n = \eta/(\pi \cdot (D_\eta/2)^2)$ (Pierre et al., 2012). With $a_f = h \cdot D_\eta$, it follows that

$$\lambda = 4\eta \frac{h}{D_\eta \pi}. \tag{35}$$

In Eq. (34), the dynamic roughness length increases with the characteristic height and density of the roughness elements. The

influence of density is assumed to saturate above a sufficiently large value. In this implementation, $\eta$ and $h$ are assumed to scale with LAI:

$$h = h_{\text{max}} \frac{\text{LAI}}{\text{LAI}_{\text{max}}}. \tag{36}$$

where $h_{\text{max}}$ is the maximum annual vegetation height and $\text{LAI}_{\text{max}}$ is the LAI above which dust emission is precluded. This approximation entails that $\eta = 1$ and $h = h_{\text{max}}$ for $\text{LAI} = \text{LAI}_{\text{max}}$, decreasing linearly until $\eta = 0$ and $h = 0$ for $\text{LAI} = 0$. Due

to the lack of data at global scale we currently assume $h_{\text{max}} = 0.4\,\text{m}$, a value obtained for the Sahel (Mougin et al., 1995; Pierre et al., 2012). We also set $\text{LAI}_{\text{max}} = 0.3$ as in the Community Land Model (Mahowald et al., 2010; Kok et al., 2014a) although this value should be further tested and constrained in future studies. We note that while $\eta$ should scale with LAI at low fractional





ground cover, the scaling may be weaker as leaves start overlapping, an aspect that is currently neglected in our simplified approach. In model grid cells, in which both $z_{0\text{stat}}$ and $z_{0\text{dyn}}$ are available, we use the larger value, $z_0 = \max(z_{0\text{stat}}, z_{0\text{dyn}})$.

The correction $f_v$ is smallest (i.e. roughness is largest) for roughness elements like stones or tall and closely spaced vegetation. Although Eq. (32) incorporates these dependencies, there is uncertainty related to characterizing the height and spacing of roughness elements, particularly where they are of irregular size or spacing. For example, in regional studies, $X$ has been set to $10\,\text{cm}$ (Marticorena and Bergametti, 1995), $40\,\text{cm}$ (Sahel; Pierre et al., 2014), and $12{,}255\,\text{cm}$ to extend its use to rougher vegetated surfaces (US; MacKinnon et al., 2004). The assumption of vegetation patches of $5\,\text{m}$ in diameter was suggested as

optimal for the Sahel (Pierre et al., 2012), but may be inadequate for other semi-arid regions. We note that this value can easily be modified in a static or dynamic way as soon as more detailed information becomes available. MONARCH uses maps of monthly LAI (actual year or climatology) and interpolates the monthly values to the day of simulation for each grid cell. The static roughness length and annual averages of the dynamic roughness length and the resulting drag partition correction, $f_{v_M}$, using $X$ from MacKinnon et al. (2004) are shown in Fig. 1 (d–f). Compared to $f_{v_R}$, the correction $f_{v_M}$ tends to be weaker

with values typically above 0.35. Areas with low corrections generally coincide with those in $f_{v_R}$, but $fv_M$ is smaller (weaker correction) in the Taklamakan and Gobi deserts and east/south-east of the Caspian Sea. When specifying $X$ according to Marticorena and Bergametti (1995) or Pierre et al. (2014) instead, the resulting drag partition is substantially more restrictive.

Apart from the effect of vegetation or other roughness elements to absorb atmospheric momentum, they also directly prohibit particle entrainment from the area they cover. Similarly, areas covered by snow/ice ($\eta_{\text{snow}}$) or bedrock ($\eta_{\text{br}}$) preclude particle

emission. We take this into account by scaling the obtained dust emission flux with $s_{\text{bare,M}} = (1 - \eta) \times (1 - \eta_{\text{snow}}) \times (1 - \eta_{\text{br}})$ in combination with the drag partition from Marticorena and Bergametti (1995) and with $s_{\text{bare,R}} = (1 - \eta_{\text{snow}}) \times (1 - \eta_{\text{br}})$ in combination with the drag partition parameterization from Raupach et al. (1993). The area covered by vegetation is already accounted for in the latter, which determines the fraction of shear stress acting on the uncovered surface (Raupach et al., 1993; Webb et al., 2020). Alternatively, the bare soil fraction can be applied to the saltation flux. Accounting for $s_{\text{bare}}$ in either the

dust emission flux or the saltation flux, but not both, assumes that saltation impacts eject dust close to their origin, i.e. saltation trajectories are short. This may not always be the case and saltating particles may also impact on the vegetated surface fraction in a grid cell, where no emission occurs.

### 3.1.5 Particle-size distribution at emission

The particle-size distribution of emitted dust is key to quantifying the emitted dust mass, dust loading in the atmosphere, dust

interactions with the energy and water cycles, along with more general impacts of dust upon climate. Whether or not the emitted dust PSD changes with the magnitude of atmospheric forces is still debated (e.g. Kok, 2011b; Shao et al., 2020). The S01, S04, S11, and KS14 dust emission schemes estimate size-resolved dust emission fluxes, the PSDs of which vary with atmospheric forcing. In contrast, the K14 scheme assumes a PSD that is independent of wind speed. The G01 scheme originally distributed the estimated bulk dust emission flux across four particle-size classes (Sec. 3.1.3) and the MB95 scheme does not include

assumptions of emitted dust particle sizes. For the latter two schemes, a pre-specified PSD is assigned to the estimated bulk dust emission flux that can be chosen to follow either D'Almeida (1987) or Kok (2011a). Figure 2 compares the PSDs based on



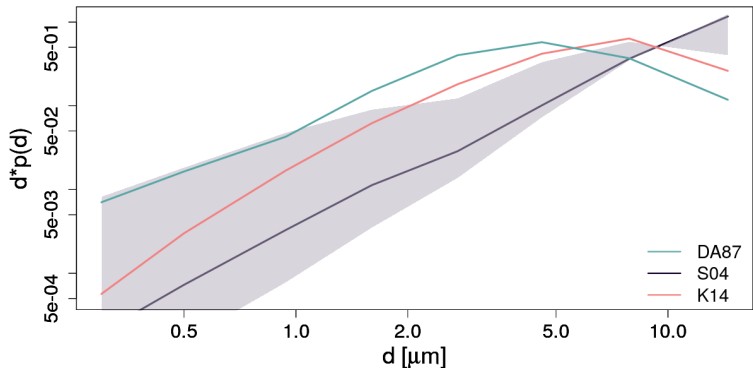

**Figure 2.** Normalized particle-size distributions (PSDs) based on D'Almeida (1987) (DA87, turquoise), Shao (2004) (S04, grey), and Kok (2011a) (as in K14, coral). The DA87 and K14 PSDs are invariant, while the S04 PSD varies in time and space. Shown for the latter are the PSDs corresponding to the 50th percentile (median; solid grey line) of the emission-weighted average diameters per model grid cell of annually accumulated dust emissions, framed by the PSDs belonging to the 5th and 95th percentiles (grey shading). The S04 PSDs were obtained from the S04-experiment presented in Sec. 4.1.

D'Almeida (1987) (DA87) , Shao (2004) (S04), and Kok (2011a) (as in K14). The K14 PSD is shifted toward coarser particle sizes compared to the DA87 PSD, indicating that the DA87 PSD describes dust after more settling of coarse constituents. Both PSDs show a peak in the diameter range 4–8 μm. The mean PSD based on S04 is continuously increasing with particle size,

however, the PSD corresponding to the 5th emission-weighted percentile of mean particle diameter with respect to annual emissions does also exhibit a peak around 8 μm, similar to the K14 PSD. In contrast, the S04-PSD belonging to the 95th weighted percentile of mean particle diameter shows a somewhat steeper increase with particle diameter, and correspondingly a smaller fraction of small particles than the median S04 PSD and the DA87 and K14 PSDs. Differences in the PSD of dust at emission yield also differences in airborne dust PSD, which has important effects on the resulting dust optical depth and

radiation interactions.

### 3.1.6 Dust sources and lower boundary conditions for emission

In MONARCH, areas from which dust emission is possible are described using a map obtained from the climatological (for the years 2003–2015) frequency of occurrence (FoO) of Moderate Resolution Imaging Spectroradiometer (MODIS) Deep Blue dust optical depth (DOD) greater than 0.2 (Hsu et al., 2004; Ginoux et al., 2012, see Sec. 4.3.1). Note that this specification of

potential dust source areas, is done in a binary sense and independent of any scaling of saltation or dust emission fluxes (see also next paragraph). This means that dust can be emitted if the topographic mask is non-zero, or the retrieved FoO(DOD>0.2) is greater than a small value (here 0.025) (Fig. 3 top panel). Areas fully covered by vegetation, snow (obtained from reanalysis data





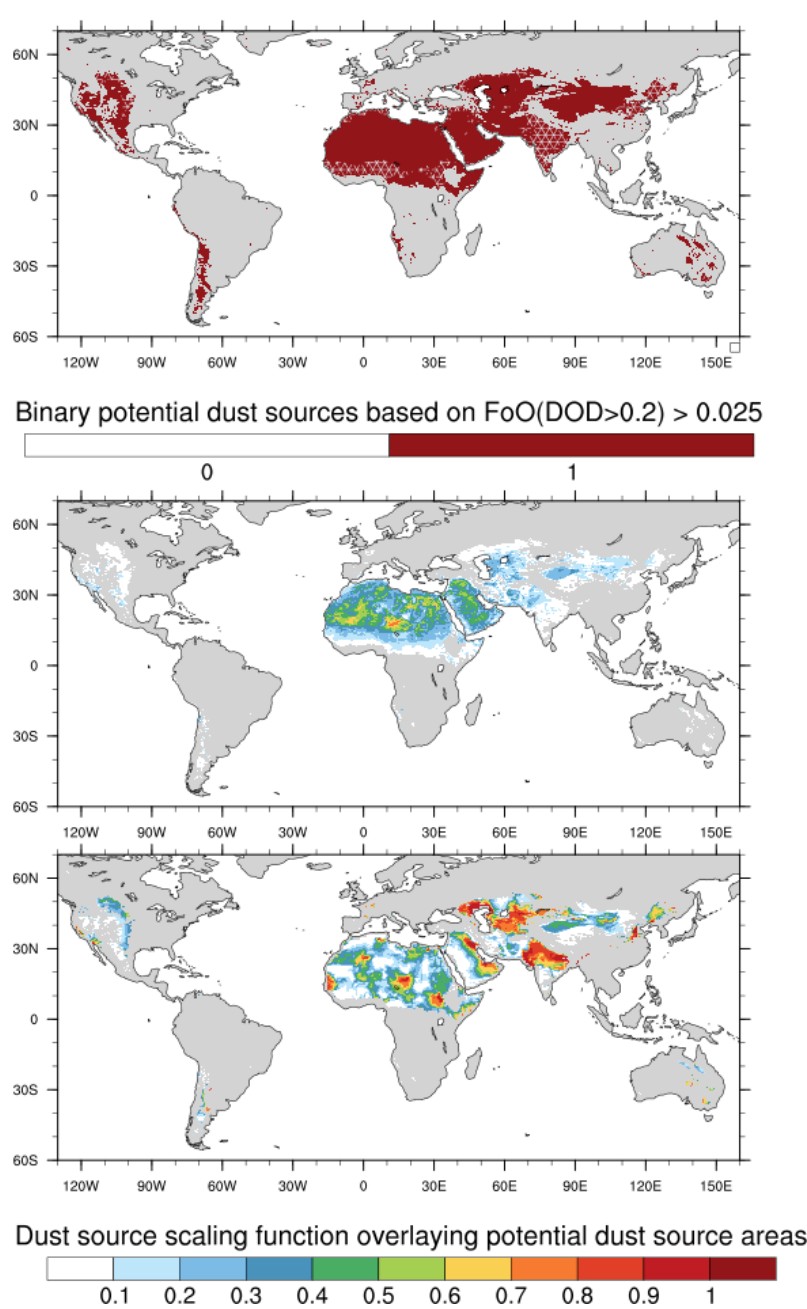

**Figure 3.** (Top) Binary potential dust source areas defined based on FoO(DOD > 0.2) > 0.025; light line patterns indicate anthropogenic dust sources using the method from Ginoux et al. (2012) considering cropland and pasture based on Klein Goldewijk et al. (2017); (Center) binary dust source overlaid with FoO(DOD> 0.2); (Bottom) binary dust sources overlaid with the topographic source scaling function from Ginoux et al. (2001) without vegetation mask.





used as boundary conditions), or bedrock (from STATSGO-FAO data) are excluded from potential dust sources as described in the final paragraph of Sec. 3.1.4, and a land-sea mask is applied.

In addition to the definition of areas from which dust emission is possible, a scaling of the calculated dust emission fluxes with the above-mentioned dust source functions is deployed in the MB95 and G01 schemes. The preferential source map from Ginoux et al. (2001) describes the sources as a function of topography. In practice, the topographic source term ($S$ in Eqs. 2 and 3) enhances dust emission from enclosed basins in arid regions where soil particles have accumulated after fluvial erosion of the surrounding highlands (Fig. 3 bottom panel). Such a scaling is part of the design of the G01 scheme and was found to

improve results compared to observations also for the MB95 implementation (Pérez et al., 2011). We have also added the option to apply the new FoO map as the preferential source function (Fig. 3 center panel). In this case, dust emission is enhanced in areas with high FoO. The purpose of a source map scaling is to compensate for unrepresented processes and surface properties, which affect dust emission. The S01-S11, K14, and KS14 schemes are not scaled with any preferential source map. For these schemes, the scheme physics is assumed to account for spatial variations in the emitted dust mass and the retrieved FoO map

is only used as a mask defining the areas from which dust emission is possible as described above.

        An additional special feature of MONARCH is its ability to tag dust originating from natural and anthropogenic (agricultural) sources. For this purpose, the MODIS FoO-based map is linked with fractions of anthropogenic land use, following the approach described in Ginoux et al. (2012), but using an updated land-use data set (Klein Goldewijk et al., 2017). When considering cropland and pasture as anthropogenic dust sources, the main anthropogenic source regions are in the Sahel, India,

China, and the United States (Fig. 3 top panel). Besides tagging natural and anthropogenic dust sources (Klose et al., 2018), MONARCH's tagging functionality can be adapted to track dust also from other predefined source origins (Kok et al., 2021).

        Vegetation in MONARCH is prescribed based on satellite data, using either an AVHRR monthly climatology of green vegetation cover fraction (Gutman and Ignatov, 2010), or monthly photosynthetic and non-photosynthetic vegetation cover based on MODIS and Landsat surface reflectance (Guerschman et al., 2015) either as a climatology or for the actual year of

simulation. The two cover fraction data sets can be used consistently within MONARCH's meteorological and dust modules. Additionally, monthly MODIS leaf-area-index (LAI) data (Myeni et al., 2015) is available for use in the dust module, for the actual year or as a climatology, in combination with the AVHRR climatological vegetation used for meteorology (evaporative fluxes).

        Soil texture class information in MONARCH is obtained from the hybrid STATSGO-FAO data set at a resolution of 30

arc seconds (0.0083°) (Pérez et al., 2011). Additional soil information, such as on soil mineral content, is currently being implemented (Gonçalves Ageitos et al., 2021b, in prep.). To aggregate soil texture data to model resolution, MONARCH utilizes a predominance approach, i.e. the predominant soil texture class in each grid cell is applied to the entire cell.

## 3.2   Dust transport and deposition

Dust transport and deposition in MONARCH has been thoroughly described in Pérez et al. (2011) and is only briefly summa-

rized in this section. The numerical schemes for dust transport by advection and turbulent diffusion are the same as those of other scalars in the NMMB model. Horizontal advection is solved with the Adams–Bashforth scheme and vertical advection





with the Crank–Nicholson scheme. Lateral diffusion follows the Smagorinsky non-linear approach. Gravitational settling of dust is solved implicitly from top to bottom using a gravitational settling velocity based on the Stokes–Cunningham approximation. As the settling velocity increasingly deviates from Stokes settling for large particles (approximately $>10\,\mu m$) and
to correct for potential numerical diffusion (Ginoux, 2003) and other unaccounted phenomena (Stout et al., 1995; van der Does et al., 2018; Dey et al., 2019), we successively reduce the settling velocity using bin-wise tuning factors. By default we use 1, 1, 1, 1, 0.5, 0.3, 0.2, 0.1 from bins 1 to 8. An explicit formulation is also now available in the model. Dry deposition through turbulent diffusion is based on Zhang et al. (2001), which accounts for Brownian diffusion, impaction, interception, and gravitational settling (Slinn, 1982). Wet deposition in MONARCH includes in-cloud and below-cloud scavenging from
both stratiform (grid-scale) and convective (sub-grid scale) clouds. In-cloud scavenging from stratiform clouds is proportional to dust mass and solubility along with the conversion rate of cloud water to rain by autoconversion, accretion, and shedding of accreted cloud water and to the conversion rate of cloud ice to precipitation through melting. Solubility is assumed to have intermediate values between purely hydrophobic and purely hydrophilic particles, with values decreasing with increasing particle size (Zakey et al., 2006). Below-cloud scavenging for rain and snow is based on Slinn (1984) and includes the effects of direc-
tional interception, inertial impaction and Brownian diffusion. For convective scavenging, the model follows the principles of the Betts–Miller–Janjic (BMJ) convective parameterization scheme developed by Betts (1986); Betts and Miller (1986); Janjic (1994). In-cloud scavenging is proportional to dust mass and solubility along with the production of precipitation in the convective cloud. Below-cloud scavenging also follows Slinn (1984) assuming a raindrop diameter of 1 mm. BMJ is a convective adjustment scheme and therefore does not represent mass fluxes. Dust is vertically mixed by deep convection in analogy with
the vertical adjustment of moisture (Pérez et al., 2011). Currently, dust particles do not affect cloud formation in MONARCH. Parameterizations representing the effect of dust particles on cloud formation, as they act as cloud condensation and ice nuclei, are planned to be implemented in the future.

### 3.3 Radiation and optical properties

The model's radiation scheme is RRTMG (Iacono et al., 2000, 2008). MONARCH allows multiple options for setting the
dust microphysical properties. In the longwave (LW), we assume refractive indices from the OPAC dataset (Hess et al., 1998) and spherical particle shape. In the shortwave (SW) we use mineralogy-based refractive indices and non-spherical shapes. The multi-component Maxwell Garnett theory (Markel, 2016) is used to calculate refractive indices of internal mixtures of 8 minerals (Gonçalves Ageitos et al., 2021a, in prep.), whose size-resolved proportions are estimated based on the mineralogical atlas from Claquin et al. (1999) combined with the brittle fragmentation theory of Kok (2011a). The single-mineral refractive
indices are taken from Scanza et al. (2015). We obtain size- and wavelength-dependent real and imaginary indices for each of the 28 soil types in the atlas and we take the median values. Note that the dependence of our refractive indices upon size is due to changes in mineralogy with size. Our median imaginary indices compare better than OPAC values (too absorbing) with recent chamber-based retrievals (Di Biagio et al., 2019), in-situ aircraft measurements (Denjean et al., 2016) and ground-based remote sensing (Balkanski et al., 2007) of dust refractive index (Gonçalves Ageitos et al., 2021a, in prep.). We account for
the effects of the substantial dust asphericity (Huang et al., 2020) on dust optics by combining the probability distributions of

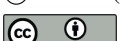



**Table 6.** Physical and optical particle properties available in MONARCH for eight particle-size bins: equivalent volume radius ($r_v$), effective radius ($r_e$), density ($\rho_p$), real and imaginary parts of the refractive index (ref$_{REAL}$, ref$_{IMAG}$), mass-extinction efficiency (MEE, [m$^2$ g$^{-1}$]), single-scattering albedo (SSA), and asymmetry factor (ASY). The optical properties are for a wavelength of 550 nm and MEE, SSA, and ASY are given assuming ellipsoidal (index ell) or spherical (index sph) particle shape. The diameter ranges of each bin are given in Sec. 3.

| Property | Bin 1 | Bin 2 | Bin 3 | Bin 4 | Bin 5 | Bin 6 | Bin 7 | Bin 8 |
|---|---|---|---|---|---|---|---|---|
| $\rho_p$ [kg m$^{-3}$] | 2500 | 2500 | 2500 | 2500 | 2650 | 2650 | 2650 | 2650 |
| $r_v$ [µm] | 0.15 | 0.25 | 0.47 | 0.80 | 1.36 | 2.29 | 3.93 | 7.24 |
| $r_e$ [µm] | 0.15 | 0.25 | 0.45 | 0.78 | 1.32 | 2.24 | 3.80 | 7.11 |
| ref$_{REAL}$ | 1.4945 | 1.4945 | 1.4945 | 1.4945 | 1.5200 | 1.5373 | 1.5442 | 1.5467 |
| ref$_{IMAG}$ | 0.0017 | 0.0017 | 0.0017 | 0.0017 | 0.0015 | 0.0014 | 0.0013 | 0.0013 |
| MEE$_{ell}$ [m$^2$ g$^{-1}$] | 1.90 | 3.24 | 2.93 | 1.55 | 0.73 | 0.41 | 0.22 | 0.11 |
| SSA$_{ell}$ | 0.98 | 0.99 | 0.99 | 0.97 | 0.95 | 0.93 | 0.90 | 0.85 |
| ASY$_{ell}$ | 0.50 | 0.71 | 0.77 | 0.75 | 0.76 | 0.81 | 0.83 | 0.85 |
| MEE$_{sph}$ [m$^2$ g$^{-1}$] | 2.27 | 3.54 | 2.21 | 0.84 | 0.49 | 0.29 | 0.16 | 0.08 |
| SSA$_{sph}$ | 0.99 | 0.99 | 0.99 | 0.96 | 0.95 | 0.93 | 0.90 | 0.84 |
| ASY$_{sph}$ | 0.59 | 0.72 | 0.70 | 0.65 | 0.75 | 0.79 | 0.81 | 0.84 |

particle shape obtained in Kok et al. (2017) based on laboratory measurements (e.g. Okada et al., 2001; Kandler et al., 2007) with the dust single-scattering database of Meng et al. (2010). Table 6 summarizes key dust properties used in MONARCH.

## 4   Model performance and evaluation

A range of global model simulations were performed with MONARCH for one year (2012) to demonstrate MONARCH's dust
modeling capabilities. We used different configurations in the runs covering different dust emission schemes. We evaluate the presented simulations against MODIS (Ginoux et al., 2012; Hsu et al., 2013) and Aerosol Robotic Network (AERONET, Giles et al. 2019) products in terms of dust optical depth (DOD).

### 4.1   Experimental setup

The global model runs performed with MONARCH were conducted at a horizontal resolution of 1°latitude × 1.4°longitude
with 48 vertical layers and a computational time step of 3 min. Turbulence, surface layer, dust emission, sedimentation and dry deposition routines were called every 4 computational times steps, moist convection, microphysics and wet scavenging routines every 8 time steps, and short- and longwave radiation routines were called every 20 time steps. The runs were initialized using ERA Interim reanalysis data (Berrisford et al., 2011; Dee et al., 2011). The meteorological fields are re-initialized daily, whereas dust fields and soil moisture are transferred between the daily runs. We used one year of spinup for soil moisture and
one month of spinup for the dust fields before the one-year simulation. A simple double-call mechanism computes the total (all

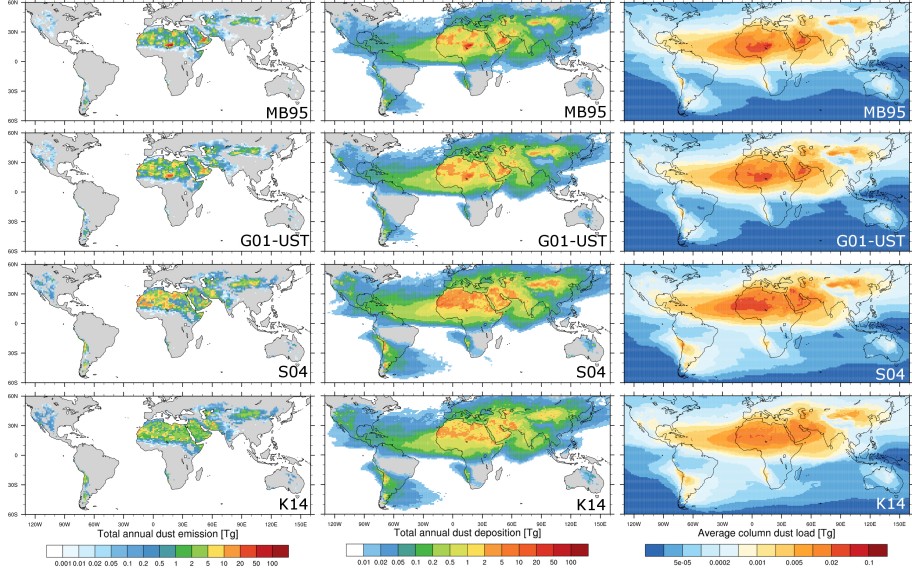

**Figure 4.** Total annual dust emission (left), dust deposition (center), and annual average column dust load for 2012 using the configurations described in Sec. 4.1. Dust deposition includes gravitational settling, turbulent diffusion, and wet deposition from convective and non-convective precipitation. Shown are results for dust particle diameters up to 20 μm.

size bins together) direct radiative effect (DRE), and a more complex multiple-call mechanism generates the DRE per size bin. The DRE per bin depends on the vertical distribution of particles in a specific bin with respect to those in other bins. Hence, the sum of the DRE per bin does not exactly equal the total DRE, especially for locations with high dust loading. To minimize errors due to such non-linearities, the DRE per bin is calculated as the difference between the total DRE with all bins included

(reference state) and the total DRE without the specific bin. Results were output three-hourly for the global runs.

Here we present results of global MONARCH simulations using the MB95, G01-UST, S04, and K14 dust emission schemes, a set of well-known and frequently used parameterizations. In all runs, we scaled soil moisture using $c_{f_1} = 0.1$ and applied the default soil moisture corrections listed in Tab. 4 (Sec. 3.1.3). We used the drag partition from Marticorena and Bergametti (1995) with $X = 12,255$ cm (MacKinnon et al., 2004) for all runs presented here. The intention of using the same drag partition

is to ease inter-comparison between the runs, and not to achieve the best possible results for each run. For the latter, different settings for each of the schemes may be more appropriate. Dust emissions in both the MB95 and G01 schemes include a scaling with the topographic source mask from Ginoux et al. (2001) shown in Fig. 3 (bottom), whereas the S04 and K14 schemes do not receive any scaling accounting for preferential dust sources.

The dust fields of all model runs were calibrated using experiment-specific global calibration factors, which were obtained

by comparing monthly averages of modeled coarse DOD (size range 1.2–20 μm) for each experiment with the DOD obtained from MODIS (see Sec. 4.3.1 for more detail) and minimizing the overall error (Cakmur et al., 2006). This calibration only





removes the general global bias for each run and does not affect the spatio-temporal variability of the dust emission, transport, deposition, and interactions.

## 4.2 Dust emission and deposition

The total mass of dust emitted globally during 2012 was 3489, 3627, 5994, and 3739 Tg, respectively, for the MB95, G01-UST, S04, and K14 dust emission schemes. Correspondingly, the total dust deposition (dry/wet) obtained with the four schemes was 3442 (2435/1007), 3541 (2131/1410), 5893 (3929/1964), and 3664 (2215/1449) Tg. Dry dust deposition here includes both gravitational settling and turbulent diffusion. Wet deposition is due to convective and non-convective precipitation. The globally integrated annual average column dust load for the four configurations resulted as 29.0, 29.1, 40.6 and 31.4 Tg. Figure 4 shows

the global spatial distribution of the total annual dust emission and deposition, as well as average column dust load for the four model runs. Values are summarized in Tab. 7.

The similarity in global dust emission between the MB95 and G01-UST schemes is a result of the scaling with the topographic source mask. Nevertheless, differences in the magnitude of dust emission are evident, in particular in the Middle East, central Asia, and Australia. Neither the S04 nor the K14 scheme uses a preferential source function besides the binary treatment

explained in Sec 3.1.6. Hence, dust emissions are independent of this source function and differences to other experiments are more pronounced. Compared with the MB95 and G01-UST runs, for example, the Bodélé Depression in Chad does not stand out as much compared to the runs using the topographic source mask. Dust emissions in Asia extend over a larger area in the S04 and K14 runs and tend to be larger in North and South America. The S04 run shows decreased dust emissions in the eastern Sahara, whereas North African and Middle Eastern dust emissions are relatively homogeneous in the K14 run. Overall,

the S04 scheme produced substantially more dust emission and deposition than the other schemes. This is a result of the on average coarser particle-size distribution in the S04 scheme above 10 μm (Fig. 2) and also reflected in the shorter lifetime of dust aerosol obtained with the S04 experiment (Tab. 7). All experiments were calibrated so that their global DOD resembled that of MODIS. The coarser particles in the S04 experiment have only a small contribution to DOD, but constitute a large amount of the emitted and deposited dust mass.

Consistent with the differences in dust emission between the four runs, the annual total dust deposition and annual average dust load are similar in the MB95 and G01-UST runs, with pronounced individual source regions such as the Bodélé Depression. In comparison, deposition and dust loading are more intense in northwestern Africa and the Middle East in the S04 scheme, and more homogeneous in the K14 scheme.

Figure A1 shows the percent contribution of dust emission and deposition at each location to their respective global and

annual totals to investigate differences between the four experiments independent from the overall flux magnitudes. The relative emission (deposition) confirms the differences highlighted before: The spatial patterns of the MB95 and G01-UST are similar due to the use of the preferential source function. In contrast, the S04 experiment produced relatively more dust in north-western Africa, while the K14 scheme generated relatively homogeneous patterns across northern Africa and the Middle East.



**Table 7.** Statistical dust parameters of four global model simulations using the dust emission schemes MB95, G01-UST, S04, and K14 with the configurations described in Sec 4.2. $\langle \text{MEE}_g \rangle$ and $\langle \text{MEE} \rangle$ are annual global averages of MEE (all sizes). $\langle \text{MEE}_g \rangle$ is calculated as the ratio of annual average grid-based DOD and dust load, whereas $\langle \text{MEE} \rangle$ is calculated from annual global average DOD and dust load. Parameters are for dust with particle diameters up to $20\,\mu\text{m}$.

| Parameter | MB95 | G01-UST | S04 | K14 |
|---|---|---|---|---|
| Total annual emission [Tg] | 3489 | 3627 | 5994 | 3739 |
| Total annual dry deposition [Tg] | 2435 | 2131 | 3929 | 2215 |
| Total annual wet deposition [Tg] | 1007 | 1410 | 1964 | 1449 |
| Total annual deposition (dry and wet) [Tg] | 3442 | 3541 | 5893 | 3664 |
| Annual average area-integrated dust load [Tg] | 29.0 | 29.1 | 40.6 | 31.4 |
| Annual average lifetime (load/deposition) [d] | 3.1 | 3.0 | 2.5 | 3.1 |
| Annual average DOD | 0.034 | 0.032 | 0.041 | 0.035 |
| Annual average $\langle \text{MEE}_g \rangle$ [$\text{m}^2\,\text{g}^{-1}$] | 1.10 | 1.11 | 1.15 | 1.01 |
| Annual average $\langle \text{MEE} \rangle$ [$\text{m}^2\,\text{g}^{-1}$] | 0.60 | 0.57 | 0.52 | 0.57 |
| Annual average SSA | 0.954 | 0.952 | 0.955 | 0.952 |

## 4.3 Dust optical depth

The global annual average of dust optical depth (DOD) is 0.034, 0.032, 0.041, and 0.035 in the MB95, G01-UST, S04, and K14 runs. This results in an average mass-extinction efficiency $\langle \text{MEE}_g \rangle$ of, respectively, 1.10, 1.11, 1.15, and $1.01\,\text{m}^2\,\text{g}^{-1}$ for the four runs considering dust up to $20\,\mu\text{m}$ in diameter (Tab. 7), calculated from grid-based annual average DOD and dust load, and, correspondingly, 0.60, 0.57, 0.52, and 0.57 based on global annual average DOD and dust load ($\langle \text{MEE} \rangle$). To provide a comprehensive, yet concise evaluation, we compare the DOD averaged across the four model runs with retrieved DOD from MODIS Deep Blue (Hsu et al., 2013; Sayer et al., 2013) and AERONET (Holben et al., 1998; O'Neill et al., 2003; Giles et al., 2019). Our objective here is to evaluate the overall behavior of MONARCH across dust emission schemes, rather than that of each individual scheme.

### 4.3.1 Comparison of modelled DOD with MODIS Deep Blue

We estimate DOD from MODIS using daily AOD and SSA at $550\,\text{nm}$, and Ångström exponent (AE) of the Deep Blue Collection 6 Level 2 MODIS products (Hsu et al., 2013; Sayer et al., 2013) from the Aqua platform at $0.1°$ resolution (Ginoux et al., 2010). As in Ginoux et al. (2012) and Pu and Ginoux (2018), DOD is estimated from AOD using a continuous function of AE (Anderson et al., 2005). Pu and Ginoux (2018) estimated an error of $\pm(0.08 + 0.52\text{DOD})$ for the DOD derived from MODIS Aqua.

To enable a direct comparison between MODIS satellite observations and MONARCH results independent of the model output frequency, MONARCH internally diagnoses the all-sky DOD for a given satellite overpass time for each day. The sampling





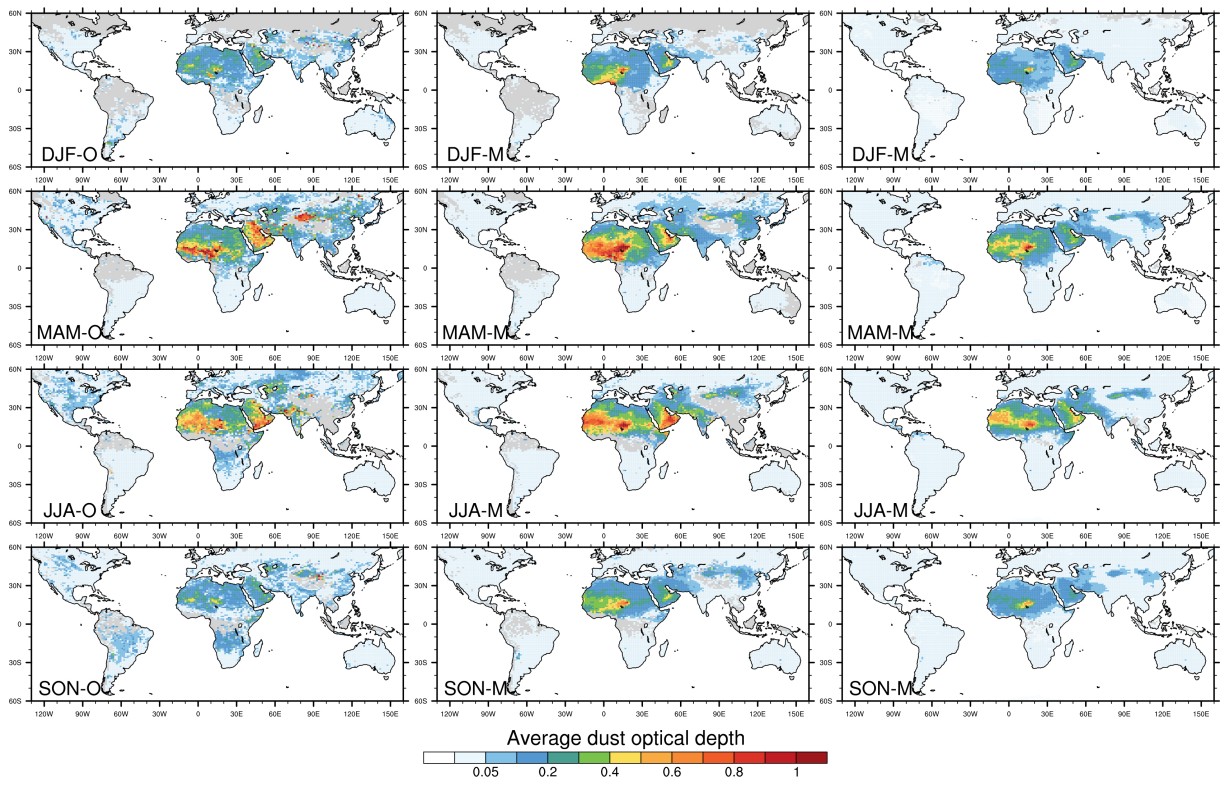

**Figure 5.** Seasonally averaged MODIS Deep Blue DOD (left), MONARCH all-sky DOD$_{coarse}$ at satellite overpass times co-located with MODIS DOD (middle), and clear-sky DOD$_{coarse}$ at approximate satellite overpass times derived from three-hourly model output from MONARCH (right). The model results were obtained for DOD$_{coarse}$ averaged across the four model experiments. The seasonal averages were calculated with respect to the number of valid values per grid cell in the respective products.

of the satellite overpass time follows Quaas (2011, http://www.euclipse.eu/downloads/D1.2_euclipse_modissimulator.pdf) and is done based on a longitude-based local time (LLT). We assume 13:30 LLT as the overpass time of MODIS Aqua. Actual overpass times vary and may deviate slightly from this nominal time. The same diagnostic is also available for MODIS Terra (nominal overpass time 10:30 LLT). Other polar satellite overpasses can be implemented easily. For model evaluation, the

MONARCH modeled satellite-DOD is additionally co-located in space and time with the satellite observations, i.e. grid cells for which the MODIS data contain missing values because of clouds are filtered from the MONARCH data for each day. MONARCH also estimates the DOD under clear-sky conditions (i.e. without clouds) based on the modeled cloud fraction and a coin-flipping method. The clear-sky DOD is currently diagnosed at the model output times (in contrast to the satellite overpass times available for the all-sky DOD). For that reason we apply a post-processing and sample the clear-sky DOD for

the output time closest to the satellite overpass time (subsequently termed "approximate satellite overpass time"). For comparison with the MODIS DOD, which discriminates coarse particles from the total AOD, we use modeled DOD in the size-range (1.2–20 μm) and refer to it as DOD$_{coarse}$.



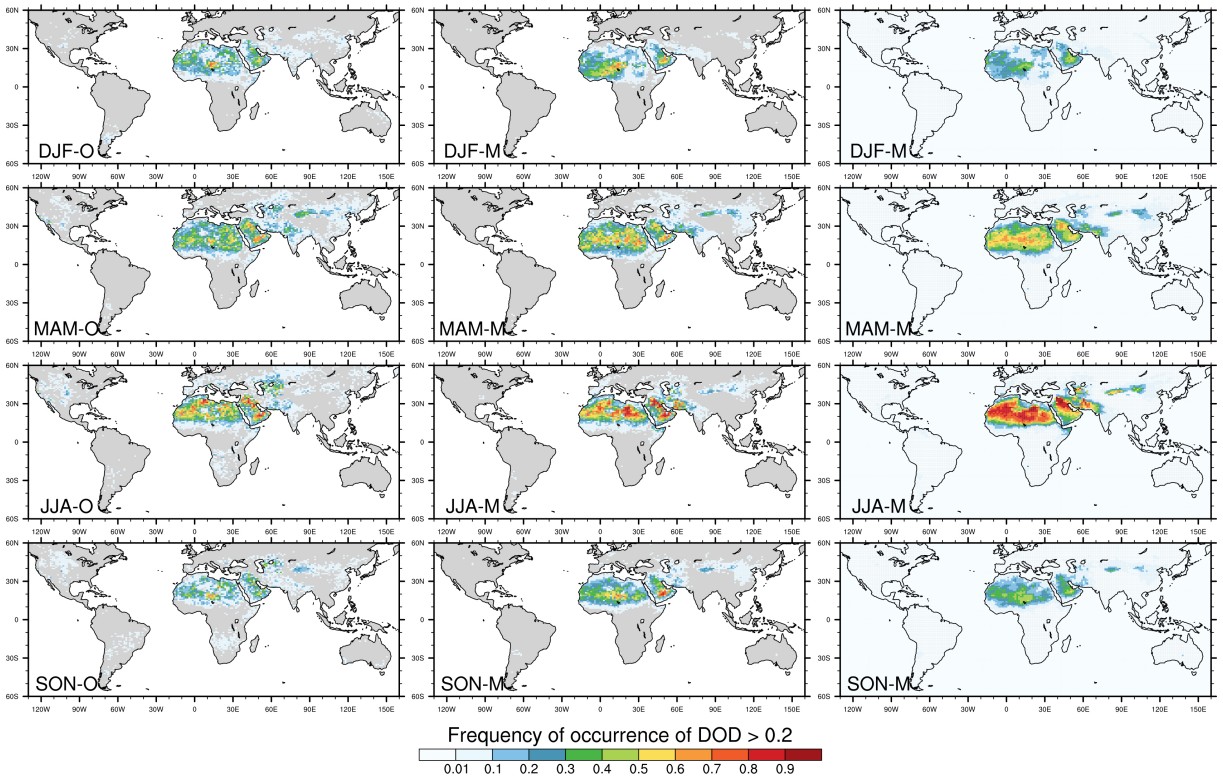

**Figure 6.** Seasonally averaged FoO of DOD > 0.2, normalized by the number of days per season for MODIS Deep Blue (left), MONARCH all-sky $DOD_{coarse}$ at satellite overpass time co-located with MODIS DOD (middle) and MONARCH clear-sky $DOD_{coarse}$ at approximated satellite overpass times derived from three-hourly model output (right). The FoO was calculated with respect to the number of days in the season.

Figure 5 shows seasonal averages of MODIS DOD (left) and modeled global all-sky co-located $DOD_{coarse}$ at satellite over-pass times (center) and clear-sky $DOD_{coarse}$ at approximate satellite overpass times (right) averaged across the four global

MONARCH runs. The spatial patterns of observed DOD in northern Africa and the Middle East are well represented in MONARCH throughout the year. Distinct features are high DOD in the Bodélé area (somewhat overestimated mainly in MAM and SON) with elevated levels also toward the south/south-west in MAM and toward the west/north-west in JJA; and increased AOD along the eastern coast of the Arabian Peninsula in MAM and in its southern part in JJA. The spatio-temporal evolution of DOD in central Asia also agrees well between MONARCH and MODIS, with relatively low values in SON and DJF and in-

creased DOD in particular in the Thar and Registan deserts in MAM and JJA. The DOD north of the Aral lake is underestimated in MONARCH compared to MODIS in JJA. Likewise, DOD in the Taklamakan desert is lower in MONARCH compared to MODIS in DJF and particularly in MAM. This may be related to the pronounced topographic features in the area, which are difficult to resemble at coarse model resolution. The DOD in Australia is relatively low throughout the year in both MODIS and MONARCH, but with areas of slightly increased DOD in north-eastern Queensland in DJF, which are underrepresented



in MONARCH. Similarly, a somewhat higher DOD in southern Africa in JJA and SON, in South America in SON and DJF, and in North America in MAM, JJA, and SON are underestimated in MONARCH. Note that the algorithm used to derive DOD from MODIS AOD cannot perfectly discriminate dust from other aerosols. This may lead to an overestimation of DOD in areas in which dust is not the dominant aerosol type and where other aerosols are present. Seasonal averages of modeled all-sky co-located $DOD_{coarse}$ for each individual model run are shown in Fig. B1.

Differences in the modeled all-sky co-located and clear-sky $DOD_{coarse}$ underline the impact of the time, location, and number of missing values on the average $DOD_{coarse}$. The clear-sky $DOD_{coarse}$ tends to be somewhat smaller compared to the all-sky co-located $DOD_{coarse}$, indicating a discrepancy between modeled and observed clouds, in combination with differences in the underlying DOD. However, the spatial patterns between both model products are overall consistent. The reduced $DOD_{coarse}$ in northern Africa matches even better with the observed DOD, whereas the modeled clear-sky $DOD_{coarse}$ in the Arabian Peninsula is smaller than in the observations. Other areas show very similar results between the all-sky co-located and clear-sky model results.

Figure 6 shows the frequency of occurrence (FoO) of DOD > 0.2, normalized by the number of days in each season, again for MODIS DOD (left), as well as modeled all-sky co-located (center) and clear-sky (right) $DOD_{coarse}$. The spatial patterns of the observed FoO are very well captured by the MONARCH runs, in particular for the all-sky co-located FoO, for key dust sources in northern Africa (e.g. Erg of Bilma/Bodélé Depression, Grand Erg Oriental/Erg Chech, El Djouf desert), the Middle East (e.g. Rub' al Khali and Nefud deserts), and central and eastern Asia (e.g. Registan/Thar and Karakum deserts, Taklamakan and Gobi deserts). Discrepancies in the all-sky co-located FoO magnitude between the model ensemble and MODIS depend on the season: The FoO is slightly underestimated in deserts East of the Caspian Sea in MAM and JJA, in South America and South Africa in JJA and SON, in Australia in SON and DJF, and in North America throughout the year. FoO values are slightly overestimated in the Arabian peninsula in JJA and SON, and eastern North Africa in MAM and JJA. The modeled seasonal all-sky co-located FoO for each individual model run is shown in Fig. B2.

The FoO obtained from the modeled clear-sky $DOD_{coarse}$ is generally larger than that obtained from the all-sky co-located $DOD_{coarse}$. Due to the normalization with the number of days in the season for calculation of the FoO, differences in the frequency and location of clouds in MONARCH and MODIS directly impact the resulting FoO. Over dust source regions, MONARCH produces considerably fewer cloud-pixels in its clear-sky product and hence a larger number of valid data values than are in the MODIS observations (not shown). As a result, the clear-sky FoO is based on a larger number of valid (high-$DOD_{coarse}$) values and is therefore larger than the MODIS and MONARCH all-sky co-located FoOs.

Fig. 7 shows global averages of monthly DOD and FoO from MODIS and the all-sky co-located $DOD_{coarse}$ from MONARCH. Globally, the DOD obtained with MODIS is reproduced well with MONARCH in all four experiments (Pearson correlation coefficient 0.98 for the experimental average; between 0.86 and 0.97 for the individual runs). The DOD range across the four experiments is relatively similar throughout the year with the spread being the largest during the northern hemispheric peak dust season in March. The MB95 experiment contributes the largest DOD in January until March and the lowest during much of the remaining year, whereas the K14 experiment shows opposite behavior. The G01 and S04 experiments are intermediate between the other two runs and best resemble the monthly global DOD for the given configurations. The correlation between



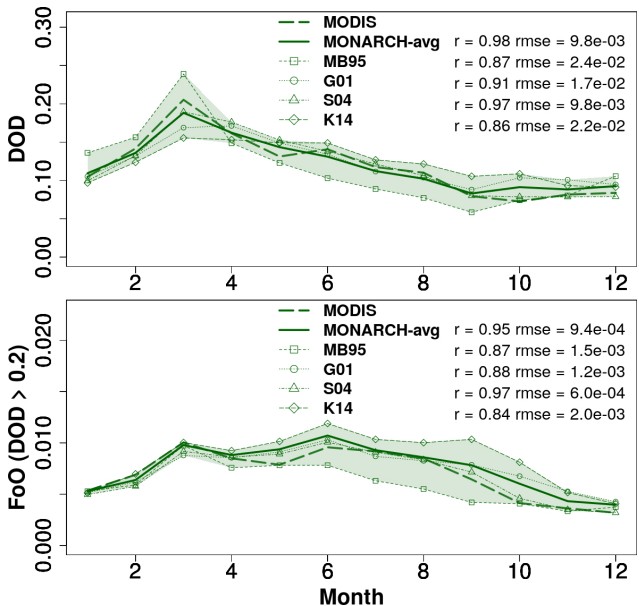

**Figure 7.** Globally averaged monthly global DOD$_{coarse}$ (top) and FoO of DOD > 0.2 (bottom) for MODIS (green dashed line) and MONARCH (DOD$_{coarse}$ all-sky co-located with MODIS observations) (green solid line). The shading indicates the range of DOD$_{coarse}$ across the four MONARCH experiments, which are also shown. Pearson correlation coefficients (r) and root-mean-squared errors (rmse) are given in the figure for both the experimental average as well as the individual runs.

the FoO of DOD > 0.2 from MODIS and from MONARCH is also very high (0.95 for the MONARCH average; between 0.84 and 0.97 for the individual runs). Whereas the results from the four experiments are very similar from November till April, the variability increases during the other months with the largest range in September. The results for the four individual runs are qualitatively similar to those for DOD, with the MB95 and K14 experiments providing, respectively, the lower and upper frames from approximately April until November, and the G01 and S04 runs being intermediate.

**4.3.2    Comparison of modelled DOD with AERONET**

AERONET is a global network of ground-based solar photometer stations (Holben et al., 1998; O'Neill et al., 2003; Giles et al., 2019). The primary parameter derived by AERONET (i.e. direct-sun) is the AOD in multiple spectral channels with uncertainties lower than 0.03. AOD data are computed for three data quality levels: level 1.0 (unscreened), level 1.5 (cloud-screened), and level 2.0 (cloud-screened and quality-assured). The products from inverting sky radiance measurements are the

aerosol size distribution, single scattering albedo, refractive index, effective radius, and asymmetry factor. AERONET has very good coverage across the globe, albeit with lower station density in remote dust source regions, such as northern Africa, the Middle East, central/western Asia, and Australia. Recently, sun–sky–lunar photometers extended the use of AERONET during nighttime (Barreto et al., 2013), allowing continuous aerosol monitoring.



**Figure 8.** Comparison of 3-hourly DOD between MONARCH (average (turquoise line) and standard deviation (shading)) and AERONET Direct Sun V3 Lev 2.0 for selected stations covering Cape Verde and the Canary Islands (Capo Verde, Santa Cruz de Tenerife), the Sahara and Sahel (Ouarzazate, Tamanrasset, Cinzana, Banizoumbou), the Middle East (Eilat, Solar Village, Masdar Institute), Asia (Karachi, Issyk-Kul, Dalanzadgad), Europe (Granada), southern Africa (Henties Bay), Australia (Birdsville), and North and South America including the Carribbean (Railroad Valley, CASLEO, Ragged Point). The direct-sun DOD is filtered for dust aerosol using AE < 0.3 (filled circles). Records which do not meet the AE criteria are less likely to be associated with dust and are shown as open circles. The Pearson correlation coefficients (corr) and root-mean-square errors (rmse) are given in each panel.



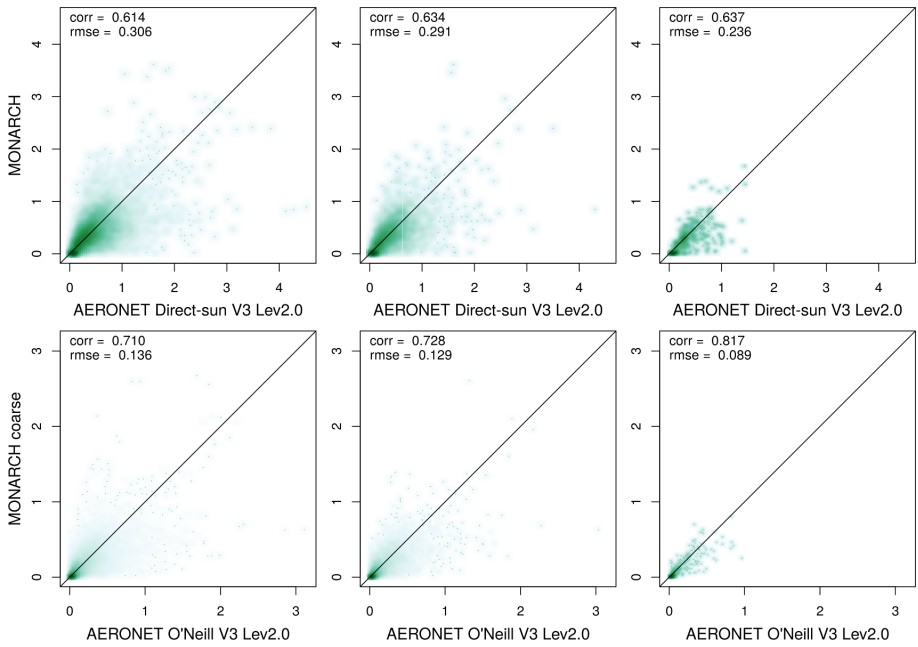

**Figure 9.** (top) Scatter plots of 3-hourly, daily, and monthly DOD estimated from AERONET direct-sun V3 Lev 2.0 (AE < 0.3) and total DOD from MONARCH (average across runs) averages for the stations listed in Appendix C; (bottom) same as (top), but for AERONET O'Neill V3 Lev 2.0 and coarse (diameters 1.2–20 µm) MONARCH DOD. The Pearson correlation coefficient (corr) and root-mean-square error (rmse) are given in the plot.

Through AOD, AERONET gives information about the aerosol content and the mode-dominant type (i.e. fine or coarse
modes) in the atmospheric column, but not the atmospheric dust burden. Almost pure mineral dust is difficult to find, except in specific areas close to desert dust sources. Instead, dust is often mixed in variable percentages with other aerosols. To isolate the atmospheric dust burden and estimate the DOD, two approaches are typically used.

The first approach aims to identify records in which the measured aerosol is dominated by mineral dust based on AE. AE is in general inversely related to the average size of the airborne particles and can be used to distinguish species with large
particles like dust and sea salt. As a rule of thumb, a larger AE indicates smaller particle size. AE is typically in the range 0–4, where the upper limit corresponds to molecular extinction, and the lower limit corresponds to coarse-mode aerosols (sea-salt and mineral dust), indicating no wavelength dependence of AOD (O'Neill et al., 2003). Since sea-salt is related to low AOD ($< 0.03$; Dubovik et al. 2002) and mainly affects coastal stations, large coarse-mode AOD values are mainly related to mineral dust. According to previous studies (Dubovik et al., 2002; Wang et al., 2004; Todd et al., 2007; Basart et al., 2009), AE values
between 0.75 and 1.2 are associated with mixed aerosols (including dust). An AE lower than 0.2–0.3 is associated with a highly dominant coarse mode in the AERONET bi-modal size distribution (Schuster et al., 2006), which corresponds to almost pure dust conditions over land. Here, we use AE < 0.3 to estimate AERONET DOD for comparison with the DOD (all sizes) obtained from MONARCH.





The second widely used methodology to estimate AERONET DOD is based on the Spectral Deconvolution Algorithm (SDA)

retrievals (O'Neill et al., 2003). The SDA algorithm estimates fine (sub-micron) and coarse (super-micron) AOD at a standard wavelength of 500 nm (AOD$_{\text{fine}}$ and AOD$_{\text{coarse}}$, respectively). Near dust source regions, DOD$_{\text{coarse}} \approx$ AOD$_{\text{coarse}}$. The advantage of this method is the availability of retrievals in regions where dust occurrence is sporadic and other aerosols are predominant, and where a more restrictive criterion, such as AE < threshold may filter out some dust intrusions (Cuevas et al., 2015). As DOD$_{\text{coarse}}$ from MONARCH, we use DOD in the diameter-range 1.2–20 μm.

For comparison with AERONET, we use bilinear interpolation to extract time series from the 3-hourly global model DOD and DOD$_{\text{coarse}}$ for the locations of AERONET measurements. We use 3-hourly averages of AERONET observations, such that a comparison with the 3-hourly instantaneous MONARCH data assumes a statistical similarity between the temporally averaged AERONET DOD and DOD$_{\text{coarse}}$ and the spatially interpolated MONARCH DOD and DOD$_{\text{coarse}}$. Figure 8 shows time-series of 3-hourly DOD from AERONET direct-sun and MONARCH for 18 selected stations in the vicinity of dust sources

around the globe: four stations in northern Africa and two stations in the typical dust outflow region west of northern Africa; three stations in the Middle East; three stations in Asia; and one station each in Europe, southern Africa, Australia, northern America, southern America, and the Caribbean. In addition, Fig. 9 compares modeled and observed DOD and DOD$_{\text{coarse}}$ for the 57 stations listed in Appendix C. The station locations are shown in Fig. C1. The time-series demonstrate an overall good agreement between the average modeled and observed DOD where the temporal variability is mostly reproduced with

discrepancies for individual DOD peaks. Consistent with MODIS DOD, the AERONET DOD tends to be small at the stations in southern Africa, Australia, and northern and southern America and AE is often not below 0.3, i.e. at least part of the DOD is likely due to aerosols other than dust. Correlations between MONARCH and AERONET are smaller for these stations, because MONARCH DOD represents pure dust and because a mismatch between individual peaks receives more weight if the number of dust episodes is small. The Pearson correlation coefficients for all other stations range between around 0.3 (Masdar Institute,

Solar Village) and 0.7 (IER Cinzana, Karachi, Ragged Point).

Taking into account the entire station list (Appendix C), the correlation is 0.61 with a root-mean-square error (rmse) of 0.31 for the total DOD and, 0.71 with an rmse of 0.14 for DOD$_{\text{coarse}}$, based on the three-hourly MONARCH data (Fig. 9). For total DOD, the correlation remains fairly constant when comparing daily and monthly instead of three-hourly values. The rmse decreases slightly with an increasing averaging period as then discrepancies for individual peaks become less relevant.

A similar behaviour is found for DOD$_{\text{coarse}}$, but with a slightly more pronounced increase also of the correlation. The overall agreement between MONARCH and AERONET is also similar across experiments (Fig. C2), however, the MB95 and S04 schemes tend to overestimate events with large DOD, whereas the G01 and K14 show an underestimation of such situations. As a result, the individual schemes show a slightly lower correlation and higher rmse than the experimental average for both DOD and DOD$_{\text{coarse}}$.

## 4.4   Direct radiative effect

Figure 10 shows the shortwave, longwave and total direct radiative effect (DRE) at the surface (SFC) and the top of the atmosphere (TOA). The longwave DRE at the SFC is positive and most pronounced in the dust belt ranging across northern





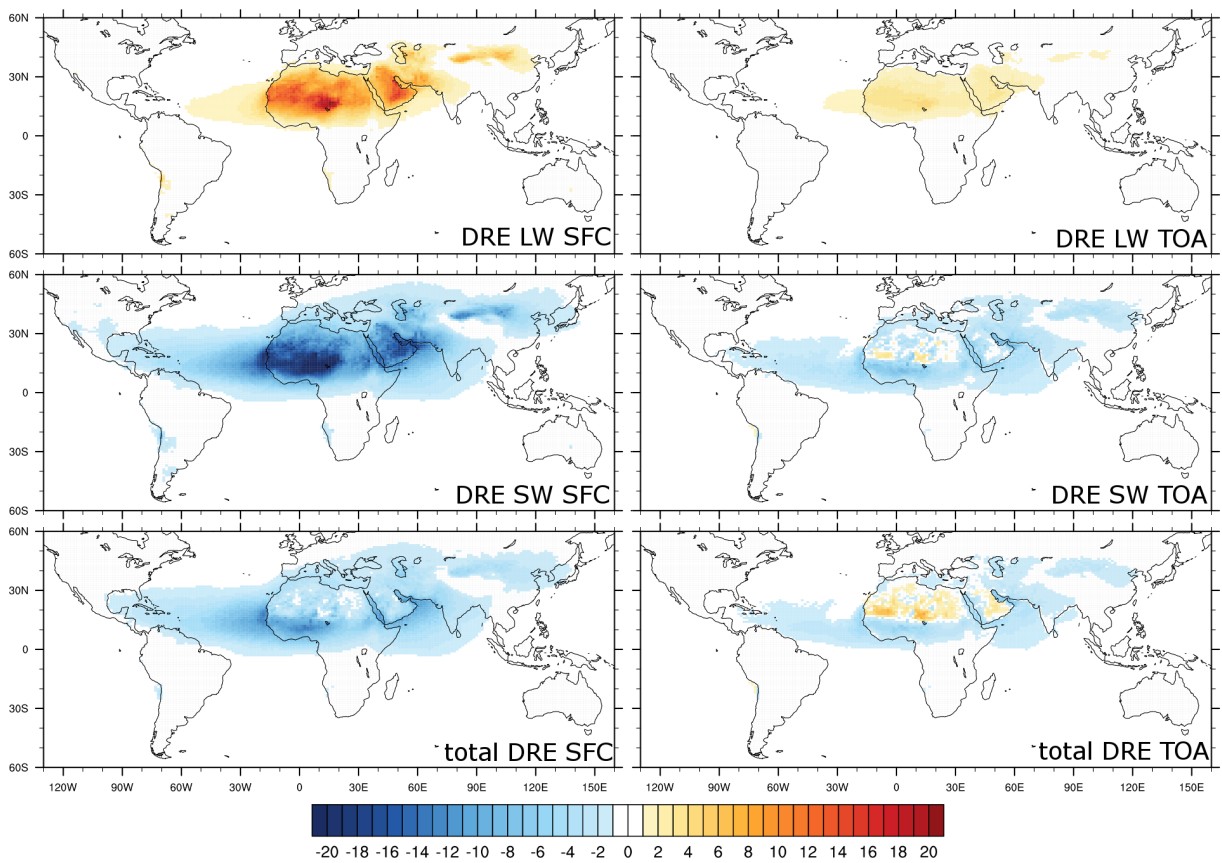

**Figure 10.** Annual average longwave, shortwave, and total direct radiative effect [W m$^{-2}$] at the surface (SFC) and the top of the atmosphere (TOA) obtained from MONARCH as average across the four runs.

Africa, the Middle East, and southwestern Asia. In the dustiest areas of northern Africa and the Middle East, the longwave SFC DRE reaches between 10 and 20 W m$^{-2}$. This strong sensitivity to the presence of dust is a result of the low atmospheric moisture content in this area. High near-surface atmospheric temperatures enhance the longwave downwelling radition (Miller et al., 2014). The longwave DRE at the TOA is smaller and typically under 5 W m$^{-2}$ due to the opposing effects of scattering and absorption by dust at the TOA. The shortwave DRE is strongly negative at the SFC, with values exceeding $-20$ W m$^{-2}$ in the main dust regions. At the TOA, the shortwave DRE is slightly negative in most areas, but slightly positive in some of the northern African dust sources related to the relatively bright underlying desert surface. This results in a negative total DRE at the SFC, with the largest (negative) values in the Sahel, the eastern Atlantic, and the Arabian Sea. At the TOA, the total DRE is positive around the main North African dust sources and slightly negative/neutral elsewhere. The globally averaged DRE at the SFC and TOA for all particle sizes as well as from each bin (relative contribution with respect to the DRE for all sizes) are summarized in Fig. 11 and listed in Tab. D1.



We note that the mineralogy-based set of refractive indices used in this work describes a more scattering dust in the shortwave
with respect to other widely used prescriptions (e.g. Patterson et al., 1977; Hess et al., 1998). At the TOA, the shortwave DRE
closely oscillates around zero over bright surfaces, such as in the Sahara and Saudi Arabia, and the total DRE does not exceed
about $10\,\mathrm{Wm}^{-2}$ in these regions. In contrast, Balkanski et al. (2007) for example obtained a total DRE at the TOA of up
to $20\,\mathrm{Wm}^{-2}$ over the Sahara when using Patterson et al. (1977) (Voltz (1973) in the longwave), and lower positive values
in this region (in agreement with our values) when using the mineralogy-based refractive indices with a hematite content of
1.5% by volume. Note that Balkanski et al. (2007) found these refractive indices to be in a better agreement with AERONET
retrievals (Dubovik et al., 2002), similar to what is found by Gonçalves Ageitos et al. (2021a, in prep.) for our refractive indices.
Moreover, Miller et al. (2006) obtained a global average total DRE at the TOA of $-0.39\,\mathrm{Wm}^{-2}$ using refractive indices from
Sinyuk et al. (2003) (Voltz (1973) in the longwave). On the other hand, Miller et al. (2014) reported a value of $0.39\,\mathrm{Wm}^{-2}$
calculated using the dust distribution from Miller et al. (2006) and refractive indices from Patterson et al. (1977). Our negative
value of $-0.24\,\mathrm{Wm}^{-2}$ is therefore again more comparable with more scattering dust as described by the refractive indices of
Sinyuk et al. (2003).

## 5    Conclusions and outlook

We presented the description of mineral dust in the Multiscale Online Non-hydrostatic AtmospheRe CHemistry model (MONARCH)
Version 2.0. MONARCH contains multiple state-of-the-art options to represent dust emissions on global and regional scales,
ranging from more simplified to more complex parameterizations based on physical processes. We tested and evaluated a
set of four global model configurations for the year 2012. Comparison with observations of dust optical depth from MODIS
and AERONET showed a good model reproduction of key features of the observed dust cycle. Global annual dust emissions
ranged between around 3,500 and 6,000 Tg. Differences in modeled dust emissions between the four configurations were
mainly driven by the dust source description (use of a preferential source mask or not) and the particle-size distributions at
emission. Dust deposition ranged between about 3,450 and 5,900 Tg in 2012 globally, yielding an average dust load of  29–
41 Tg. The smaller range of simulated load among experiments is due to the shorter lifetime of the coarse particles included
in the S04 scheme that exhibits larger emission. The total direct radiative effect obtained from the MONARCH simulations is
slightly negative at the surface in dust transport regions. At the top of the atmosphere, the total direct radiative effect is positive
near the main North African dust sources and slightly negative/neutral elsewhere.
The multifaceted options of MONARCH and its dust component, combined with an advanced work flow management
for use in high-performance computing environments, makes it a powerful and versatile tool applicable for process studies,
operational forecasting, and climate research. In the following, we outline a few ongoing activities related to the MONARCH
dust component to demonstrate its capabilities.
Dust ensemble runs can be generated with MONARCH by utilizing the diverse model configurations and by perturbing
model parameters related to, for example, surface winds, soil humidity, and the spatial distribution of dust emission, which are
deemed to be uncertain. In Di Tomaso et al. (2017), perturbations were applied to the threshold friction velocity and the dust



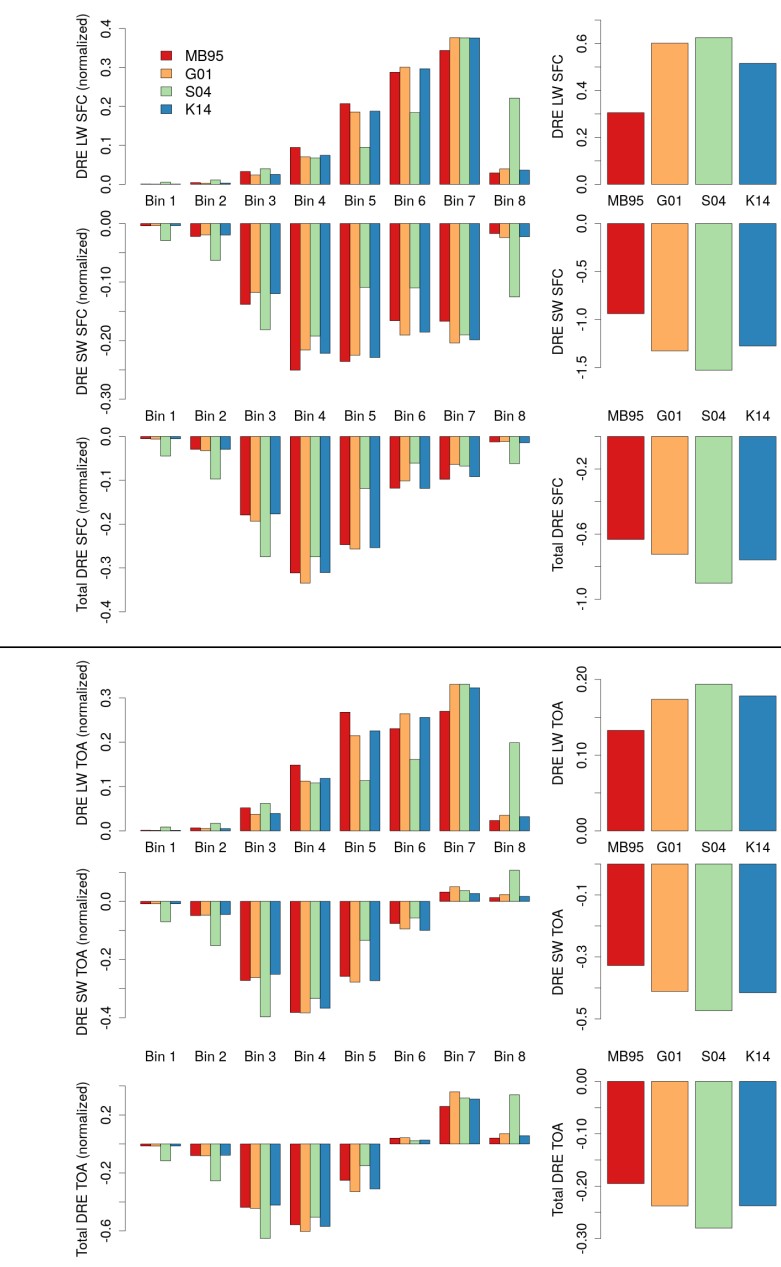

**Figure 11.** Global averages of MONARCH-derived DRE [W m$^{-2}$] for each run at the SFC and TOA for shortwave and longwave radiation and the total (shortwave and longwave). Shown are the relative contributions per bin, normalized with the absolute value of the DRE for all bins (left) and the DRE for all particle sizes (right). The diameter ranges and effective radii of each bin are given in, respectively, Sec. 3 and Tab. 6. The DRE results are also given in Appendix D.





emission flux per size bin. In Escribano et al. (2021, in prep.), different dust emission schemes – or linear combinations thereof – are used by different ensemble members. Combined meteorology and emission perturbations were shown to be necessary to produce sufficient ensemble spread in (dust) aerosol outflow regions (Rubin et al., 2016). This can be achieved using different

meteorological fields as initial and boundary conditions in the meteorological driver of MONARCH (NMMB) for each forecast run in the ensemble, in addition to the dust perturbations. In the dust reanalysis currently in production at the BSC, we use an ensemble based on stochastic perturbations of emission parameters, in conjunction with multi-physics emission schemes and multi-meteorological initial and boundary conditions (Di Tomaso et al., 2021, in prep.).

Airborne dust is not a homogeneous entity, but a mixture of minerals, the relative amounts of which depend on the source

region. Mineralogy affects a variety of dust-related impacts, e.g. interaction with radiation, atmospheric chemistry or nutrient supply to certain ecosystems. The capability to explicitly represent dust composition was recently added to MONARCH allowing the tagging of up to 12 different minerals. This new feature is currently used to assess the relevance of dust mineralogy for dust impacts and to provide insights for the near-term atmospheric and climate modeling communities (Gonçalves Ageitos et al., 2021b, in prep.).

The combination of different vegetation input data sets, drag partition approaches, and the source tagging capability, allows to represent the seasonal vegetation dynamics and provides an ideal basis to investigate the importance of dust from anthropogenic (agricultural) sources, for which a key driver is the seasonal vegetation growth and decay. The benefit of online estimates within a modeling framework is that not only the emission, but also the transport, deposition, and effect of anthropogenic dust can be investigated (Klose et al., 2018).

*Code availability.* Access to the model code is currently restricted to institutes and collaborators involved in the model development. Confidential access to the code can be granted for editors and reviewers. The model version presented in this paper is available upon request to the corresponding authors.

*Data availability.* MONARCH output presented in this paper will be made available via a public repository upon acceptance.

**Appendix A: Dust emission and deposition**

Figure A1 shows the percent contribution of dust emission and deposition at each location to their respective global and annual totals to visualize regional differences between the different experiments independent from the overall emission (deposition) magnitudes obtained.



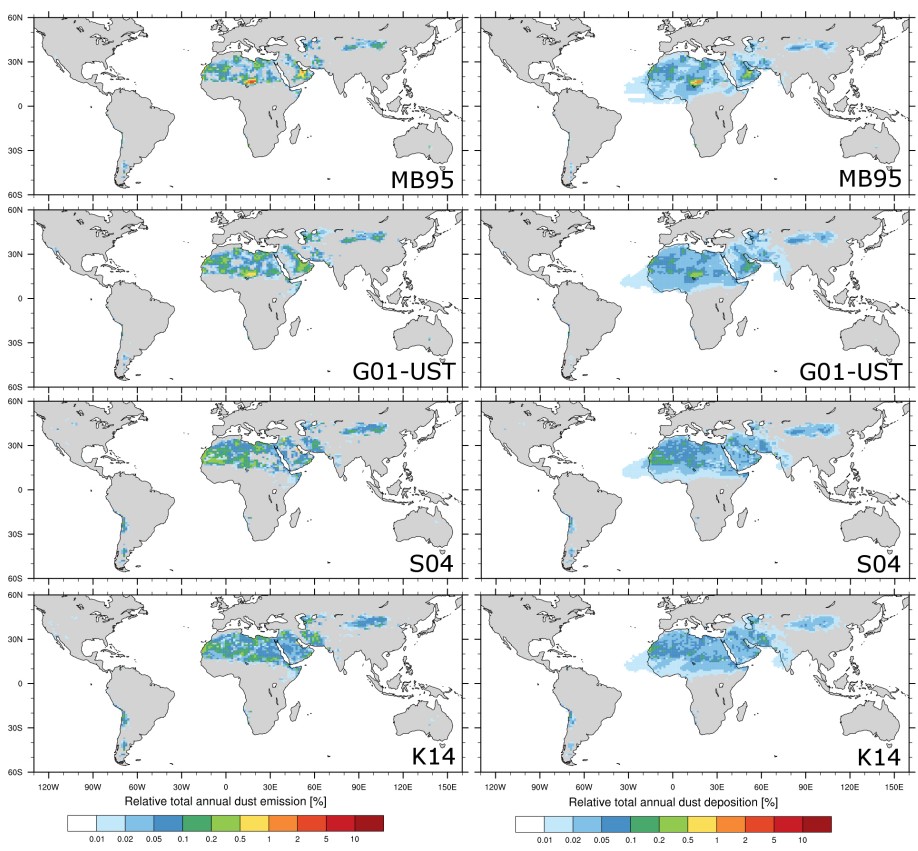

**Figure A1.** Percent contribution of dust emission (left column) and deposition (right column) to their respective global and annual totals for the four experiments described in Section 4.1.

## Appendix B: Dust optical depth and its frequency of occurrence

Figures B1 and B2 show the DOD and FoO of DOD$> 0.2$ for MODIS and MONARCH (all-sky DOD$_{coarse}$ at satellite overpass time co-located with MODIS) for the four MONARCH experiments using the MB95, G01-UST, S04, and K14 dust emission schemes.

## Appendix C: Comparison with AERONET

The AERONET stations used for comparison with MONARCH and to obtain the global calibration factor are listed in Tab. C1 and shown in Fig. C1. They cover the main dust source regions around the globe. The intention of using only a subset of all stations is to increase confidence in that aerosol detected by AERONET photometers is predominantly dust.



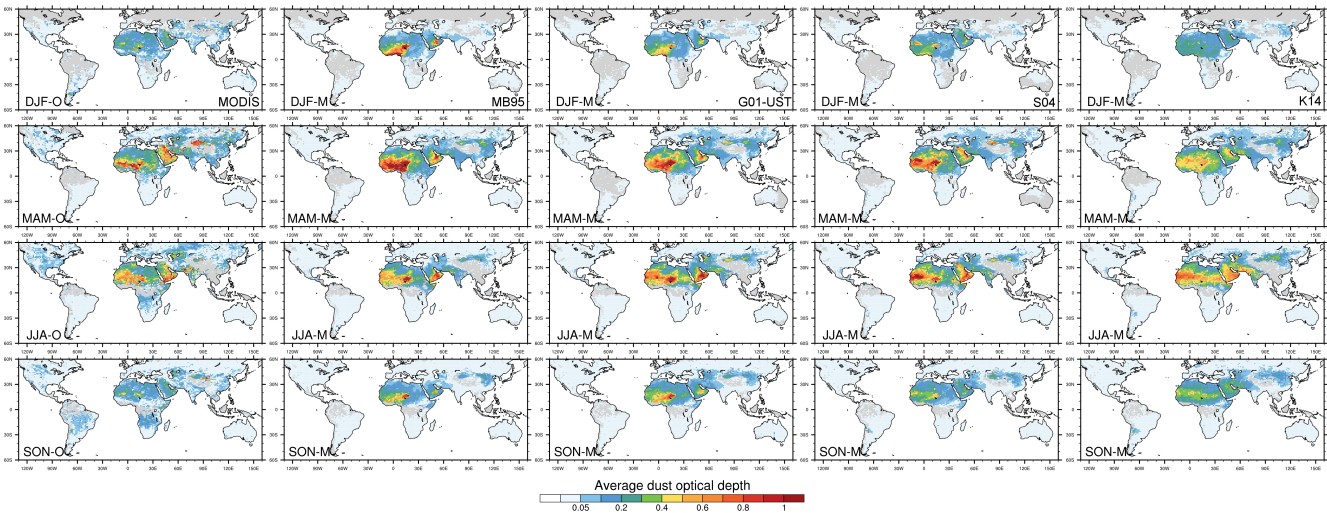

**Figure B1.** Seasonally averaged MODIS Deep Blue DOD (left) and MONARCH all-sky DOD$_{coarse}$ at satellite overpass times co-located with MODIS DOD for the MB95, G01-UST, S04, and K14 runs. The seasonal averages were calculated with respect to the number of valid values per grid cell in the MODIS product.

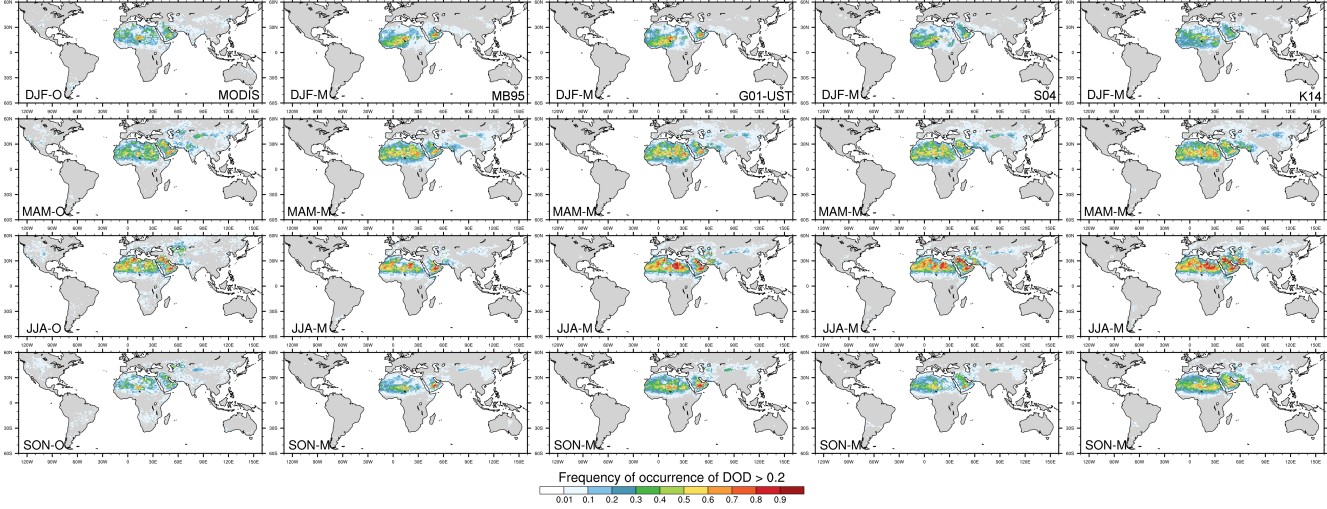

**Figure B2.** Seasonally averaged FoO of DOD > 0.2, normalized by the number of days per season for MODIS Deep Blue (left) and MONARCH all-sky DOD$_{coarse}$ at satellite overpass times co-located with MODIS DOD for the MB95, G01-UST, S04, and K14 runs. The FoO was calculated with respect to the number of days in the season.

Figure C2 compares 3-hourly DOD estimated from AERONET direct-sun V3 Lev 2.0 (AE < 0.3) and AERONET O'Neill V3 Lev 2.0 with, respectively, 3-hourly total DOD and DOD$_{coarse}$ from MONARCH for all four experiments (MB95, G01, S04, and K14) taking into account the stations listed in Tab. C1.





**Table C1.** List of AERONET stations used for comparison with MONARCH results

| Station name | Latitude | Longitude | Station name | Latitude | Longitude |
|---|---|---|---|---|---|
| Autilla | 42.00 | −4.60 | Kanpur | 26.51 | 80.23 |
| Banizoumbou | 13.54 | 2.66 | Karachi | 24.87 | 67.03 |
| Birdsville | −25.90 | 139.35 | KAUST Campus | 22.30 | 39.10 |
| Blida | 36.51 | 2.88 | La Laguna | 28.48 | −16.32 |
| Calhau | 16.86 | −24.87 | La Parguera | 17.97 | −67.05 |
| Camaguey | 21.42 | −77.85 | Lahore | 31.54 | 74.32 |
| Cape San Juan | 18.38 | −65.62 | Lampedusa | 35.52 | 12.63 |
| Capo Verde | 16.73 | −22.94 | Lecce University | 40.34 | 18.11 |
| CASLEO | −31.80 | −69.31 | Masdar Institute | 24.44 | 54.62 |
| CEILAP-Bariloche | −41.15 | −71.16 | Mezaira | 23.15 | 53.78 |
| CUT-TEPAK | 34.67 | 33.04 | Nes Ziona | 31.92 | 34.79 |
| Dakar | 14.39 | −16.96 | Ouarzazate | 30.93 | −6.91 |
| Dalanzadgad | 43.58 | 104.42 | Oujda | 34.65 | −1.90 |
| Dunhuang LZU | 40.49 | 94.96 | Ragged Point | 13.16 | −59.43 |
| Dushanbe | 38.55 | 68.86 | Railroad Valley | 38.50 | −115.96 |
| Eilat | 29.50 | 34.92 | Red Mountain Pass | 37.91 | −107.72 |
| ETNA | 37.61 | 15.02 | Saada | 31.63 | −8.16 |
| Evora | 38.57 | −7.91 | SACOL | 35.95 | 104.14 |
| FORTH CRETE | 35.33 | 25.28 | SAGRES | 37.05 | −8.87 |
| Frenchman Flat | 36.81 | −115.93 | Santa Cruz Tenerife | 28.47 | −16.25 |
| Granada | 37.16 | −3.61 | SEDE BOKER | 30.85 | 34.78 |
| Guadeloup | 16.33 | −61.50 | Solar Village | 24.91 | 46.40 |
| Gwangju GIST | 35.23 | 126.84 | Tabernas PSA-DLR | 37.09 | −2.36 |
| Henties Bay | −22.10 | 14.26 | Taihu | 31.42 | 120.21 |
| IASBS | 36.71 | 48.51 | Tamanrasset INM | 22.79 | 5.53 |
| IER Cinzana | 13.28 | −5.93 | Tizi Ouzou | 36.70 | 4.06 |
| Ilorin | 8.32 | 4.34 | Trelew | −43.25 | −65.31 |
| Issyk-Kul | 42.62 | 76.98 | White Sands | 32.92 | −106.35 |
| Jaipur | 26.91 | 75.81 | | | |

## Appendix D: Size-dependent dust direct radiative effect

Table D1 gives the relative contributions of each particle-size bin to the global average DRE for each MONARCH run.



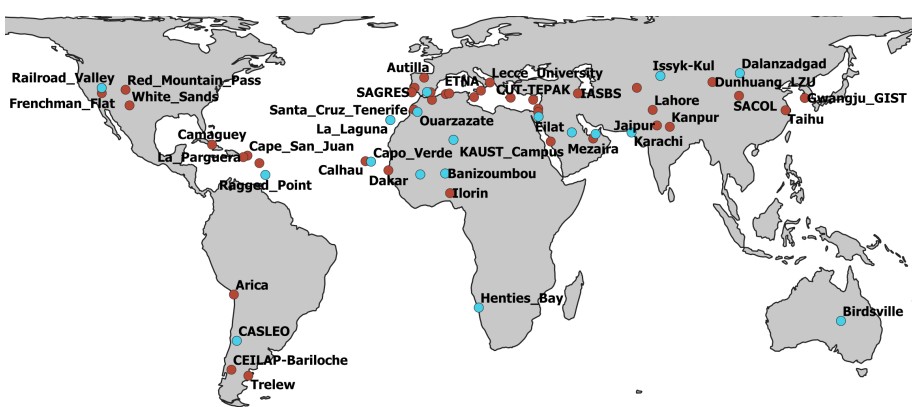

**Figure C1.** AERONET stations (direct-sun V3 Lev 2.0) available in 2012. The station subset used for comparison with MONARCH are shown in turquoise, whereas all other stations are marked in red.

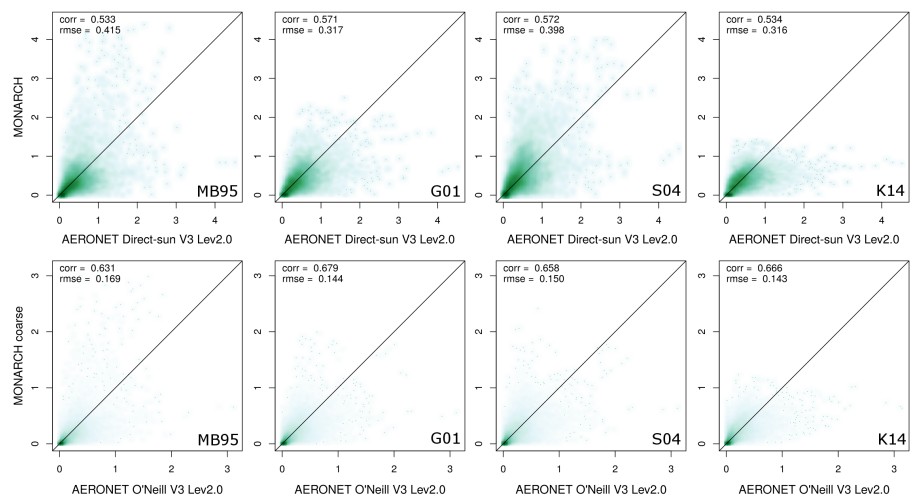

**Figure C2.** (top) Scatter plots of 3-hourly DOD estimated from AERONET direct-sun V3 Lev 2.0 (AE < 0.3) and total DOD from MONARCH by experiment (MB95, G01, S04, and K14) for the stations listed in Appendix C; (bottom) same as (top), but for AERONET O'Neill V3 Lev 2.0 and coarse (diameters 1.2–20 µm) MONARCH DOD. The Pearson correlation coefficient (corr) and root-mean-square error (rmse) are given in the plot.





Table D1: Global averages of MONARCH-derived DRE [W m$^{-2}$] (all particle sizes and relative contribution per bin) for each run at the SFC and TOA for shortwave and longwave radiation and the total (shortwave and longwave). Results are visualized in Fig. 11. The diameter ranges and effective radii of each bin are given in, respectively, Sec. 3 and Tab. 6.

| Run | MB95 | G01-UST | S04 | K14 |
|---|---|---|---|---|
| DRE LW SFC (all sizes) | 0.3053 | 0.6019 | 0.6251 | 0.5156 |
| Bin 1 | 0.0002 | 0.0003 | 0.0034 | 0.0003 |
| Bin 2 | 0.0013 | 0.0018 | 0.0068 | 0.0016 |
| Bin 3 | 0.0100 | 0.0144 | 0.0250 | 0.0130 |
| Bin 4 | 0.0289 | 0.0423 | 0.0421 | 0.0384 |
| Bin 5 | 0.0632 | 0.1117 | 0.0593 | 0.0968 |
| Bin 6 | 0.0878 | 0.1810 | 0.1149 | 0.1529 |
| Bin 7 | 0.1049 | 0.2265 | 0.2350 | 0.1937 |
| Bin 8 | 0.0089 | 0.0237 | 0.1383 | 0.0187 |
| DRE SW SFC (all sizes) | −0.9383 | −1.3264 | −1.5264 | −1.2744 |
| Bin 1 | −0.0036 | −0.0046 | −0.0443 | −0.0045 |
| Bin 2 | −0.0207 | −0.0256 | −0.0959 | −0.0249 |
| Bin 3 | −0.1295 | −0.1559 | −0.2767 | −0.1525 |
| Bin 4 | −0.2349 | −0.2866 | −0.2934 | −0.2825 |
| Bin 5 | −0.2210 | −0.2983 | −0.1664 | −0.2917 |
| Bin 6 | −0.1556 | −0.2528 | −0.1676 | −0.2363 |
| Bin 7 | −0.1566 | −0.2705 | −0.2903 | −0.2531 |
| Bin 8 | −0.0160 | −0.0317 | −0.1913 | −0.0287 |
| DRE TTL SFC (all sizes) | −0.6330 | −0.7246 | −0.9013 | −0.7587 |
| Bin 1 | −0.0032 | −0.0042 | −0.0402 | −0.0039 |
| Bin 2 | −0.0185 | −0.0235 | −0.0876 | −0.0222 |
| Bin 3 | −0.1134 | −0.1401 | −0.2475 | −0.1340 |
| Bin 4 | −0.1971 | −0.2424 | −0.2475 | −0.2356 |
| Bin 5 | −0.1562 | −0.1861 | −0.1066 | −0.1925 |
| Bin 6 | −0.0747 | −0.0734 | −0.0549 | −0.0898 |
| Bin 7 | −0.0618 | −0.0462 | −0.0608 | −0.0696 |
| Bin 8 | −0.0078 | −0.0082 | −0.0558 | −0.0107 |



Table D1 – *Continued from previous page*

| Run | MB95 | G01-UST | S04 | K14 |
|-----|------|---------|-----|-----|
| DRE LW TOA (all sizes) | 0.1327 | 0.1737 | 0.1936 | 0.1782 |
| Bin 1 | 0.0002 | 0.0002 | 0.0017 | 0.0002 |
| Bin 2 | 0.0009 | 0.0009 | 0.0033 | 0.0009 |
| Bin 3 | 0.0069 | 0.0065 | 0.0120 | 0.0070 |
| Bin 4 | 0.0197 | 0.0195 | 0.0209 | 0.0211 |
| Bin 5 | 0.0355 | 0.0373 | 0.0219 | 0.0402 |
| Bin 6 | 0.0306 | 0.0459 | 0.0312 | 0.0456 |
| Bin 7 | 0.0358 | 0.0574 | 0.0640 | 0.0575 |
| Bin 8 | 0.0031 | 0.0061 | 0.0385 | 0.0057 |
| DRE SW TOA (all sizes) | $-0.3277$ | $-0.4114$ | $-0.4735$ | $-0.4156$ |
| Bin 1 | $-0.0028$ | $-0.0035$ | $-0.0331$ | $-0.0034$ |
| Bin 2 | $-0.0160$ | $-0.0195$ | $-0.0720$ | $-0.0188$ |
| Bin 3 | $-0.0891$ | $-0.1078$ | $-0.1880$ | $-0.1042$ |
| Bin 4 | $-0.1252$ | $-0.1577$ | $-0.1580$ | $-0.1527$ |
| Bin 5 | $-0.0845$ | $-0.1142$ | $-0.0635$ | $-0.1133$ |
| Bin 6 | $-0.0249$ | $-0.0389$ | $-0.0269$ | $-0.0414$ |
| Bin 7 | 0.0105 | 0.0207 | 0.0175 | 0.0112 |
| Bin 8 | 0.0043 | 0.0096 | 0.0506 | 0.0071 |
| DRE TTL TOA (all sizes) | $-0.1950$ | $-0.2377$ | $-0.2799$ | $-0.2374$ |
| Bin 1 | $-0.0027$ | $-0.0035$ | $-0.0326$ | $-0.0033$ |
| Bin 2 | $-0.0157$ | $-0.0195$ | $-0.0713$ | $-0.0185$ |
| Bin 3 | $-0.0853$ | $-0.1056$ | $-0.1824$ | $-0.1002$ |
| Bin 4 | $-0.1087$ | $-0.1434$ | $-0.1415$ | $-0.1350$ |
| Bin 5 | $-0.0489$ | $-0.0782$ | $-0.0420$ | $-0.0737$ |
| Bin 6 | 0.0078 | 0.0102 | 0.0063 | 0.0063 |
| Bin 7 | 0.0505 | 0.0854 | 0.0886 | 0.0735 |
| Bin 8 | 0.0079 | 0.0168 | 0.0949 | 0.0135 |





*Author contributions.* MK implemented most of the dust-module upgrades, ran, analyzed, and evaluated the simulations, and wrote the manuscript. OJ contributed to the model upgrade and leads the MONARCH development together with CPGP. MGA, JE, MD, VO, and EDT contributed to the model upgrades. SB contributed to the model evaluation. GMP and FM supported the development of MONARCH and MONARCH workflow. PG, JG and CPr contributed data used in the model/for model evaluation. YH and JK contributed dust optical
properties used in MONARCH. CPGP and RM co-designed the model changes together with MK. In addition, CPGP implemented part of the model upgrades, and contributed to manuscript writing. All authors commented on the manuscript.

*Competing interests.* The authors declare that they have no competing interests.

*Acknowledgements.* MK and JE have received funding from the European Union's Horizon 2020 research and innovation programme under the Marie Skłodowska-Curie grant agreements no. 789630 (MK) and H2020-MSCA-COFUND-2016-754433 (JE). MK acknowledges fund-
ing through the Helmholtz Association's Initiative and Networking Fund (grant agreement no. VH-NG-1533). BSC co-authors acknowledge funding from the European Research Council (grant no. 773051; FRAGMENT), the AXA Research Fund, the Spanish Ministry of Science, Innovation and Universities (CGL2017-88911-R), the EU H2020 project FORCES (grant no. 821205), the CMUG-CCI3-TECHPROP contract, an activity carried out under a programme of, and funded by, the European Space Agency (ESA), and from the DustClim project, which is part of ERA4CS, an ERA-NET initiated by JPI Climate, and funded by FORMAS (SE), DLR (DE), BMWFW (AT), IFD (DK), MINECO
(ES), ANR (FR) with co-funding by the European Union (Grant 690462). BSC co-authors also acknowledge PRACE (eFRAGMENT2) and RES (AECT-2019-3-0001 and AECT-2020-1-0018) for awarding access to MareNostrum at the Barcelona Supercomputing Center and for providing technical support. RLM is funded by the NASA Modeling, Analysis and Prediction Program (NNG14HH42I). JFK acknowledges support from the National Science Foundation (NSF) grants 1552519 and 1856389 and the Army Research Office under Cooperative Agreement Number W911NF-20-2-0150 awarded. YH acknowledges NASA grant 80NSSC19K1346, awarded under the Future Investigators in
NASA Earth and Space Science and Technology (FINESST) program. The authors thank all the Principal Investigators and their staff for establishing and maintaining the AERONET sites, and the MODIS mission scientists and associated NASA personnel for the production of the AOD, SSA, and AE data used in this study.





**List of symbols**

$\alpha$      coefficient in soil moisture correction from Klose et al. (2014)

$\alpha_N$      functional parameter in KS14 scheme $[\mathrm{m}^{-2}]$

$\alpha_q$      vertical-to-horizontal-flux ratio

$\beta$      coefficient in soil moisture correction from Klose et al. (2014)

$\beta_R$      coefficient in drag partition correction from Raupach et al. (1993)

$\beta_R$      ratio of roughness-element to surface drag coefficients

$\eta$      vegetation cover fraction

$\eta_{\mathrm{br}}$      bedrock cover fraction

$\eta_{\mathrm{snow}}$      snow/ice cover fraction

$\eta_{ci}$      aggregated dust fraction at diameter $d_i$

$\eta_{fi}$      total dust fraction at diameter $d_i$

$\eta_{mi}$      free dust fraction at diameter $d_i$

$\gamma$      function in S04 scheme (Eq. 5)

$\kappa$      coefficient in S01 and S04 schemes

$\lambda$      roughness density (frontal area index)

$\Omega$      soil volume removed by a saltation impact

$\psi_*$      function in K14 scheme

$\psi_s$      saturation capillary pressure head [m]

$\rho_a$      air density $[\mathrm{kg\,m}^{-3}]$

$\rho_b$      soil bulk density (approximately $1000\,\mathrm{kg\,m}^{-3}$)

$\rho_l$      water density $[\mathrm{kg\,m}^{-3}]$

$\rho_p$      Particle density $[\mathrm{kg\,m}^{-3}]$

$\rho_{a0}$      reference air density; $\rho_{a0} = 1.225\,\mathrm{kg\,m}^{-3}$

$\rho_{bd}$      bulk density of dry soil $[\mathrm{kg\,m}^{-3}]$

$\rho_{pa}$      average soil particle density; $\rho_{pa} = 2500\,\mathrm{kg\,m}^{-3}$

$\rho_{ps}$      saltator density $[\mathrm{kg\,m}^{-3}]$

$\sigma_m$      saltation bombardment efficiency; $\sigma_m = m_\Omega / m_{p_s}$

$\sigma_v$      ratio of roughness-element basal to frontal area

$\sigma_{p_i}$      $\eta_{mi}/\eta_{fi}$

$\tau$      total stress

$\tau_s''$      maximum surface stress on exposed area

$\tau_s'$      average surface stress on exposed area

$\theta$      volumetric soil moisture content $[\mathrm{m}^3\,\mathrm{m}^{-3}]$



| | |
|---|---|
| $\theta_r$ | volumetric air-dry residual soil moisture content [$\mathrm{m^3\,m^{-3}}$] |
| $\theta_s$ | volumetric saturation soil moisture content [$\mathrm{m^3\,m^{-3}}$] |
| $\xi$ | particle resting angle |
| $a$ | coefficient in soil moisture correction from Fécan et al. (1999) |
| $b$ | coefficient in soil moisture correction from Fécan et al. (1999) |
| $C_\alpha$ | coefficient in K14 scheme |
| $c_\lambda$ | coefficient |
| $C_{\mathrm{G01}}$ | dimensional factor in G01 scheme |
| $c_{\mathrm{thr}}$ | scaling coefficient |
| $C_e$ | coefficient in K14 scheme, dependent on $u_{*st}$ |
| $c_Q$ | coefficient |
| $c_y$ | coefficient in S01, S04, and S11 schemes |
| $c_{f_1}$ | soil moisture scaling factor |
| $c_{f_2}$ | soil moisture scaling factor |
| $D_\eta$ | Diameter of vegetation patches |
| $d_{\mathrm{max}}$ | upper diameter limit of dust in MONARCH; currently $d_{\mathrm{max}} =20\,\mathrm{\mu m}$ |
| $d_c$ | coarse particle diameter in drag partition of Marticorena and Bergametti (1995) |
| $d_i$ | dust-particle diameter |
| $d_s$ | saltator diameter |
| $D_v$ | viscous sublayer depth [m] |
| $d_{i0}$, $d_{i1}$ | lower and upper diameter limits of the particle-size bin corresponding to $d_i$ |
| $F$ | vertical dust emission flux |
| $f$ | lifting force [N] |
| $f_{\mathrm{clay}}$ | clay fraction |
| $f_i$ | interparticle cohesive force [N] |
| $f_t$ | particle retarding forces [N] |
| $f_v$ | drag partition correction |
| $f_w$ | soil moisture correction factor |
| $f_{i_c}$ | capillary cohesive force |
| $f_{w_{B\mathrm{wet}}}$ | constant in soil moisture correction from Belly (1964) |
| $g$ | gravitational acceleration [$\mathrm{m\,s^{-2}}$] |
| $h$ | vegetation height |
| $h_{\mathrm{max}}$ | maximum annual vegetation height (relevant for dust emission) for use in drag partition |
| $h_w$ | function in soil moisture correction from Klose et al. (2014) |
| $L$ | Obukhov length [m] |





| | | |
|---|---|---|
| $m$ | coefficient in drag partition correction from Raupach et al. (1993) | |
| $m_\Omega$ | soil mass removed by a saltation impact; $m_\Omega = \rho_b \Omega$ | |
| $M_{\text{sand}}$ | sand fraction | |
| $m_p$ | particle mass | |
| $m_{p_s}$ | mass of particles with diameter $d_s$ | |
| $n$ | number of vegetation patches | |
| $N_p$ | particle number concentration [m$^{-3}$] | |
| $P$ | soil plastic pressure [Pa] | |
| $p$ | probability density function of $f$ [N$^{-1}$] | |
| $p_A$ | area particle size distribution [m$^{-1}$] | |
| $p_f$ | fully dispersed particle-size distribution [m$^{-1}$] | |
| $p_i$ | probability density function of $f_i$ [N$^{-1}$] | |
| $p_m$ | minimally dispersed particle-size distribution [m$^{-1}$] | |
| $p_s$ | airborne sediment particle size distribution in the S11 scheme [m$^{-1}$] | |
| $Q$ | horizontal saltation flux | |
| $Q_s$ | saltation flux of a particle with diameter $d_s$ | |
| $r_e$ | Effective radius [m] | |
| $r_v$ | Equivalent volume radius [m] | |
| $S$ | dust source scaling function | |
| $s_{\text{bare}}$ | bare soil fraction | |
| $s_p$ | particle-size fractions in G01 scheme | |
| $T_p$ | particle response time [s] | |
| $u_*$ | friction velocity [m s$^{-1}$] | |
| $u_{10\text{m}}$ | 10 m wind speed [m s$^{-1}$] | |
| $u_t$ | threshold wind speed [m s$^{-1}$] | |
| $u_{*\text{NMMB}}$ | friction velocity provided by the atmospheric model | |
| $u_{*st0}$ | minimum standardized threshold friction velocity; $u_{*st0} \approx 0.16$ m s$^{-1}$ | |
| $u_{*st}$ | standardized threshold friction velocity; $u_{*st} = u_{*t}\sqrt{\rho_a/\rho_{a0}}$ [m s$^{-1}$] | |
| $u_{*t0}$ | threshold friction velocity for dry conditions | |
| $u_{*t_{\text{dry}}}$ | model threshold friction velocity for dry conditions (including optional scaling) | |
| $u_{*t_{d0}}$ | minimum dry threshold friction velocity | |
| $u_{*t}$ | threshold friction velocity [m s$^{-1}$] | |
| $u_{t_{d0}}$ | minimum dry threshold wind velocity | |
| $w$ | gravimetric soil moisture content [%] | |
| $w_*$ | convective scaling velocity [m s$^{-1}$] | |





$w_r$       gravimetric air-dry residual soil moisture content [%]

$w_t$       particle terminal velocity [$\mathrm{m\,s^{-1}}$]

$X$         parameter in drag partition from Marticorena and Bergametti (1995)

$z_0$       aerodynamic roughness length

$z_{0\mathrm{dyn}}$   dynamic aerodynamic roughness length

$z_{0\mathrm{stat}}$  static aerodynamic roughness length

$z_{0s}$     smooth aerodynamic roughness length





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
