# Peer review of "Mineral dust cycle in the Multiscale Online Nonhydrostatic AtmospheRe CHemistry model (MONARCH) Version 2.0"

_Geoscientific Model Development, 2021_

## Author Comment (AC1)

We thank the editors for handling our submission and the reviewers for their thorough assessment of our manuscript and for their constructive comments, which we have considered carefully. We respond to each comment below.

**Reviewer 1**

**General comments**

This manuscript describes the dust module in the Multiscale Online Non-hydrostatic AtmospheRe CHemistry model (MONARCH) Version 2.0, a state-of-the-art global chemical weather prediction system. The dust module includes a variety of dust emission processes, from conventional, simplified schemes of Marticorena and Bergametti (1995) and Ginoux et al. (2001) to more physically based methods of Shao (2001, 2004), Shao et al. (2011), Kok et al. (2014), and Klose et al. (2014). The processes of the model are well documented, especially for the dust emission processes.

**Thank you very much. We greatly appreciate your positive feedback.**

In the experiment and results, the authors conducted a one-year simulation with the four configurations. The model generally shows good representations of the global dust distribution. The evaluation methods are sound, and the presentations are of good quality. However, it is desirable to have a more in-depth evaluation of the characteristics of each dust module (I am curious that which one (or the ensemble of the four ) is the default setting of the dust module for the dust forecasting). Most of the evaluations are global picture of dust emission, deposition, loading and AOD. I would like to see the evaluation of the representation of dust storm events and surface dust concentrations in a future study.

Thank you for this comment and suggestion. The developments presented in this manuscript were finalized after the last update of the model version used for the dust forecasts available from the websites of the Barcelona Dust Forecast Center (https://dust.aemet.es/) and the WMO SDS-WAS NAMEE Regional Center (https://sds-was.aemet.es/), and are therefore not yet included in the forecasting model version. We intend to include developments presented here in a future update. We agree that in-depth evaluations of specific aspects of the dust cycle as well as case studies of individual events are interesting and desirable. However, given the scope of the paper (presenting comprehensive technical developments of the dust module in MONARCH) and its length, we believe that a general/global evaluation as currently presented is most suitable and we prefer to leave additional more detailed evaluations for future studies.

Also, I think it is desirable for a multi-year simulation, not just a specific year (2012), to evaluate the model results since the dust aerosol exhibits year-to-year variability. The authors evaluate the radiative effects of dust aerosol and show the direct radiative effects of the dust. I am not sure that the dust direct radiative forcing interactively affects the meteorological process, but the radiative feedback to the meteorology should be minor because the meteorological fields are re-initialized daily by the ERA reanalysis. I think the evaluation of dust radiative effect can be a reference for the climate models: hence the multi-year evaluation of it is also desirable (in a future study). In all, I find this manuscript is suitable for publication in the Geoscientific Model Development after a minor revision.

**Thank you for this comment. Our focus here was on the intra-annual variability, the locations of dust sources and spatial patterns of dust emission and dust loading, etc., and differences thereof for**

different dust emission schemes based on our technical developments. We agree that the dust cycle shows year-to-year variability and we plan to study its representation with MONARCH with multiyear simulations in the future as suggested by the reviewer.

As correctly pointed out by the reviewer, the direct radiative effect of the dust aerosol does impact on the meteorology in our simulations, but impacts are small due to the daily re-initialization with reanalysis. In combination with the multiple-call DRE calculations, this allows to differentiate the effect of the aerosol from that of the meteorology to a large extent.

Specific (minor) comments

Line 254: How is the fixed minimal threshold (utd0 = 5ms-1) determined?

We chose 5 m s-1 as a typical lower limit for dust emission under favorable land-surface conditions. This value is in agreement with results from studies using synoptic measurements (Kurosaki and Mikami, 2007) or remote sensing and reanalysis data (Pu et al., 2020). We have added this information in Section 3.1.3.

Line 266: How the authors set the "optional constant scaling parameter" cthr ?

Cthr is a global tuning parameter, which can be set depending on model resolution, representations of surface roughness and soil moisture, etc. For our global model setup for example, we have run tests with cthr = 0.4. The effect of Cthr (<=1) is to allow dust emission for lower friction velocities and as a consequence from wider areas with the goal to minimize the global error compared to observations.

Line 638: "Fig. 7 shows ...": "Fig." should be spelled out as "Figure" at the top of the sentence.

**Corrected, thanks.**

Line 710: A typo: "radition" should be "radiation"

Thanks very much. Corrected.

Line 745: "work flow" -> "workflow"

We have changed this as suggested.

**Reviewer 2**

This manuscript offers a comprehensive description of how dust modelling is treated in the MONARCH model developed at the Barcelona Supercomputing Center.

The part that treats dust emission flux is very instructive and carefully drafted, it contains all the necessary details for any group to carefully think at how to treat emissions of dust. It is therefore a useful contribution.

After a rather complete model's description, the authors chose the year 2012 to evaluate the simulated dust against satellite and Aeronet retrieved dust optical depth (DOD). The analysis displays both time series and frequencies of occurrences of dust events for which the DOD exceeds 0.2 at 550nm. For both aspects the model performs rather well compared to other models I am aware of.

Reading the whole paper requires a certain effort since the description is rather complete but the authors were up to the task write it well. In its actual form and with addressing a few minor comments listed below, I find this work worthy of being published in GMD.

**Thank you very much for this positive feedback, which we very much appreciate.**

Major comment: Aside from the different uses of MONARCH (different scales + assimilation) the authors do not dwell for very long as of why no less than seven different emission schemes can and are used with MONARCH. I encourage them to reflect on it and do a better job at explaining to the reader what guided them to code and adapt such an unusual number of emission schemes.

Thank you for this important comment. We actually believe that the large number of conceptually different dust emission schemes is a key feature of MONARCH and we appreciate the opportunity to expand on our motivation, which we have also added to Section 3.1:

Modeling dust emission is a grand challenge due to the difference in scales between the dust emission process and model resolution, heterogeneity of the land-surface and its representation, turbulent wind forcing, and not least the complexity of the dust emission process itself comprising particle-particle and particle-fluid interactions. This is particularly problematic given that mineral dust is a dominant contributor to the atmospheric aerosol mass. The different dust emission parameterizations available all have strengths and weaknesses and by nature none of them can predict dust emission perfectly. In comparison with observations, discrepancies between results obtained with those parameterization schemes can provide insights in aspects of parameterizing dust emission, which are particularly uncertain (or not) and which may need more attention in future research. Scheme comparisons are often done indirectly by comparing results of models that utilize different parameterizations. Any discrepancies obtained in such comparisons, will include also differences in the meteorological modeling, land-surface conditions, etc., besides the dust emission parameterization. By including a variety of dust emission schemes in MONARCH, we have the unique opportunity to directly compare dust emission schemes in the exact same framework of a global model.

In addition, MONARCH is used for operational forecasting at global and regional scales and to create dust reanalysis datasets. In that context, ensemble members with different emission schemes and configurations provide a strong added value to characterize uncertainty, for example, in the context of data assimilation (Escribano et al., 2021, ACPD, in review).

We note that because we are not presenting results obtained with the aerodynamic dust entrainment scheme from Klose et al. (2014), we have decided to remove discussions on this scheme from the present paper and source code and instead present it in a future study.

Minor comments:

You should mention in the abstract that the particle size distribution (PSD) considered in this work extends to 20 mm. By the same token, you could indicate that considering particles larges particles will slightly render more positive both the SW and LW radiative effect of dust.

**Thank you for this suggestion. We have included information on the particle size range in the abstract. We included a brief statement about a presumably more positive DRE when including larger particles sizes in the main text (Sec. 4.4).**

Page 3, Il 61-73: you should add that simple formulations of dust emissions are unable to be predictive for past periods (mid-Holocene, LGM, rapid transitions...) or for future climate conditions

**Thank you for this remark. We totally agree that modeling dust emissions of past or future periods requires a dust emission scheme that is sensitive to changes in climate. At the same time, specifying accurate input data for such periods is even more challenging than for the present day. We have added a corresponding sentence.**

Page 4 line 88, I wrote in my copy of the ms: 'Why do you have such a plethoric choice of numerical schemes in MONARCH' this is link to my major comment above.

**Please see our response to your corresponding major comment above.**

Page 5, ll 122-124: 'The effective and volume radii of each bin in the radiative and sedimentation schemes respectively (see Sec. 3.3, Tab. 6) are time-invariant and based on a lognormal distribution with mass median diameter of 2.524  $\mu$ m and geometric standard deviation of 2 (Schulz et al., 1998; Zender et al., 2003).' This has implications for your results. What I mean is that you probably underestimate the presence of large particles compared to PSDs such as were measured by Ryder et al. (2013 & 2018). You should mention it.

Our model includes 8 size bins with the volume and effective radii shown in Table 6. By design, sectional models have to assume a constant sub-bin distribution. Such a sub-bin distribution can be weighed towards dust PSDs typically found over sources or PSDs that are more representative of long-range transport conditions. We agree with the reviewer our assumption may underestimate the effective and volume radii of the coarse bins compared to PSDs measured by Ryder et al. (2013, 2018). However, we have estimated that the underestimation is very low. By doubling the median radius of the lognormal distribution assumed (5.048  $\mu$ m instead of 2.524  $\mu$ m) the volume radii of the coarsest bin increases by barely 3%.

Page 8, I 195: you need a reference regarding the Brittle theory to backup this statement: 'The K14 dust emission scheme uses the concept of the fragmentation of brittle material.'

**Thanks for this suggestion. We have added references (Kolmogorov, 1941; Åström, 2006).**

Page 11 line 254: How did you come up with chosing the value of 5 m s-1 for utd0? (another reviewer had the same comment)

Thank you for this question. As responded to Reviewer 1's comments, we chose 5 m s-1 as a typical lower limit for dust emission under favorable land-surface conditions. This value is in agreement with

**results from studies using synoptic measurements (Kurosaki and Mikami, 2007) or remote sensing and reanalysis data (Pu et al., 2020). We have added this information in Section 3.1.3.**

Page 11 line 260: change 'the tail of the wind speed distribution' with 'the tail of the upper end of the wind speed distribution. Another good illustration of that is the reference Timmreck and Schulz (2014) that I include below.

**Thank you for the suggestion. Similar to what you proposed, we have changed "tail of the wind speed distribution" into "strong-wind tail of the wind speed distribution" and we have added the reference you proposed.**

Bottom of page 11: You present several schemes for moisture correction but you do not discuss their relative merits. Are they observations that would help us favor one scheme rather than another?

Thank you for your question. Whereas the soil moisture corrections are of different complexity, similar to the dust emission schemes, it is unfortunately difficult to evaluate them, because both the modeling and the measurement of soil moisture at the very surface, where dust emission happens, is challenging. Measurements of soil moisture typically happens at a few centimeters depth at best and are representative of a certain soil volume. Similarly, the top soil layer in models is also usually a few centimeters thick. At such a depth, the soil may still be moist while the soil-atmosphere interface may have already been dried by wind, permitting dust emission. For that reason, we prefer not to discuss the benefits of the different soil moisture corrections, as it would be relatively speculative.

Paragraph 3.1.5: I had the remark: Why not use the PSD from Ryder et al. (2013 & 2018)?

To approximate the PSD of dust at emission for those schemes that do not internally calculate the PSD, we need PSD data collected near the surface and as close as possible to the dust source and during active dust emission. The PSDs obtained from Ryder et al. (2013, 2018) were obtained from aircraft measurements and are therefore likely considerably different to those at the surface.

Page 19 Figure 3: Some dust specialists argue that Iceland is an important dust source, and having been in Iceland I have witnessed dust uplift there. Why is Iceland not showing up on this map? Is the frequency of events much too small over Iceland?

We agree that Iceland is a dust source and dust emissions from Iceland are generally possible in MONARCH, depending on the vegetation data set used. Due to data gaps in our implementation of MODIS LAI related to, e.g., cloud contamination, usage of MODIS LAI as in the presented experiments, does not permit dust emission from high latitudes such as Iceland. In contrast, gaps in the photosynthetic and non-photosynthetic vegetation data set also available in MONARCH have been filled as much as possible using temporal averages, hence dust emission from high latitudes is possible when this option is chosen and occurs depending on the configuration (not presented in this paper). We are working on an improved implementation of MODIS LAI, to facilitate high-latitude dust modeling with both options.

Page 20 line 449: I find the formulation: 'For these schemes, the scheme physics ...' rather odd, why not write more simply: 'The physics of these schemes...'

**Thanks for pointing this out. We have revised the sentence.**

Page 21 lines 480-482: Indicate from which measurements you inferred the solubility of dust?

Solubility in this context is defined as the fraction of tracer contained in cloud water, which can eventually be precipitated. The values are taken from Zakey et al. (2006) and represent an intermediate hypothesis between purely hydrophobic or purely hydrophilic assumptions. The values decrease with the bin size as the small particles are assumed more likely to form cloud condensation nuclei.

Page 32, line 535: you do note explain the low cutoff size for dust size range that you use (1.2 mm). Please explain.

The SDA product (O'Neill et al., 2003) computes AOD fine and coarse based on the spectral curvature of the retrieved AODs and assuming bimodal aerosol size distributions. As pointed out in O'Neill et al. (2003), there is no strict size cut at a certain radius. For comparison with models, we used a radius of 0.6  $\mu$ m established in the AERONET Inversion product (Dubovik and King, 2000), which at the same time corresponds precisely to the 5 coarsest bins of our model without needing additional assumptions.

Page 24 last paragraph: a suggestion for future work: the relative patterns of dust emission sources could be challenged with the work of Kok et al (2021) that you coauthored.

Thank you for this comment. You are right in that some of our results differ to the inverse modeling results obtained by Kok et al. (2021a,b), for example, North African dust emissions are larger (range 55-71 % compared to 50% estimated by Kok et al., 2021b), which suggests that we may underestimate the contribution from other source regions. This is most likely a result of larger vegetation/roughness element coverage in, for example, North America, which acts to suppress dust emission to a large extent in the coarse global model grids we are using. However, our model results also agree with many of the aspects highlighted by Kok et al. (2021a,b): With 3489 – 5994 Tg, our global dust emission fluxes are in the same range; Asian dust sources contribute to about 37-42% to global emissions in three of four configurations, South America about 3-4% in two configurations, Middle Eastern and Central Asian sources about 28-33% in three configurations, East Asian sources > 9% in one configuration, western North African sources contribute 6% more than eastern North African sources in one configuration. We see considerable differences between our four configurations, which again highlights the great benefit of having multiple dust options available in one model to allow for a direct comparison. It is important to note also that the model version we used for our contributions to Kok et al. (2021a,b) does not yet include all developments presented here, and that in the present paper, we chose to use the same drag partition in our four configurations to allow for an easier comparison between schemes, whereas a different configuration may have led to even better results for some schemes.

We included a table listing the contributions of the different sources to global dust emissions in Appendix A and added a discussion to Sec. 4.2.

Page 25, Table 7: I am uneasy with the large to very large differences between MEEg and MEE, do you have a simple explanation as to why you can have such large differences?

The difference between MEEg and MEE is a result of the different impacts of locations with and without dust in the two averaging methods. In the first case, MEEg is first calculated from DOD and dust load for each grid cell and then averaged, which puts equal emphasis on grid cells with and without dust. In the second case, MEE is calculated from globally averaged DOD and dust load and hence focuses on dusty locations. We have added a corresponding explanation in the text.

Page 26, line 595-596: as above we have no explanation for this choice of lower boundary of 1.2 mm of the PSD.

**Please see our response above.**

Page 27 Figure 6. If you present the following statistics: bias, rmse, correlation for the center and right column, you would be more quantitative than the paragraph of qualitative comparison you wrote. You do present statistics in Fig. 7 which is good.

**Thank you for this suggestion. We have now added area-weighted root-mean-square-errors and pattern correlations to Figs. 6, 7, B1, and B2.**

Figure 10: it would help to have the global mean values of each DRE (LW SFC, LW TOA, SW SFC, SW TOA, total SFC and total TOA) inserted on each panel of this Figure.

**We have now included the global average values in the figure. Thanks for the suggestion.**

Page 34, lines 724 and 728: You misspelled Volz. Please change 'Voltz' to 'Volz'.

**Corrected. Thanks.**

Page 36 line 857: I would rephrase 'multi-physics emission schemes' to 'emission schemes with different complexity in the physics'

**We revised the sentence. Thank you.**

References:

Ryder, C. L., Highwood, E. J., Rosenberg, P. D., Trembath, J., Brooke, J. K., Bart, M., Dean, A., Crosier, J., Dorsey, J., Brindley, H., Banks, J., Marsham, J. H., McQuaid, J. B., Sodemann, H., and Washington, R.: Optical properties of Saharan dust aerosol and contribution from the coarse mode as measured during the Fennec 2011 aircraft campaign, 13, 303–325, 3, 2013., Atmos Chem Phys, 13(3), 303–325, https://doi.org/10.5194, 2013.

Ryder, C. L., Marenco, F., Brooke, J. K., Estelles, V., Cotton, R., Formenti, P., McQuaid, J. B., Price, H. C., Liu, D., Ausset, P., Rosenberg, P. D., Taylor, J. W., Choularton, T., Bower, K., Coe, H., Gallagher, M., Crosier, J., Lloyd, G., Highwood, E. J. and Murray, B. J.: Coarse-mode mineral dust size distributions, composition and optical properties from AER-D aircraft measurements over the tropical eastern Atlantic, Atmospheric Chem. Phys., 18(23), 17225–17257, https://doi.org/10.5194/acp-18-17225-2018, 2018. Timmreck, C., and M. Schulz (2004), Significant dust simulation differences in nudged and climatological operation mode of the AGCM ECHAM, J. Geophys. Res., 109, D13202, doi:10.1029/2003JD004381.

**Executive Editor**

Dear authors,

We have checked the 'Code and Data Availability' section of your manuscript, and in its current form, it does not comply with our policy. We can understand that you have restrictions imposed by your research institution that prevent you from publishing your code; however, we need evidence. Also, this information must be in your manuscript. Therefore, (a) at least you must provide information about how to obtain a license for the code (something like a landing page for the model, more or less to prove the license restriction), (b) you must provide us with the relevant information about the restrictions that prevent you of publishing the MONARCH code and its license, and (c) you have to identify the correct version of the code that you use in the "Code and Data Availability" section. We need this information to continue with the review process, and ideally, you should post it before the end of the open discussion stage.

Best regards,

Juan A. Añel

Geosc. Mod. Dev. Executive Editor

Dear Mr. Juan A. Añel,

Compliance of our submission with Geosci. Mod. Dev.'s 'Code and Data Availability' policy has already been checked and approved before publication of our manuscript in Geosci. Mod. Dev. Discuss. as usual. We were therefore surprised to receive your message, in particular as a comment in the interactive discussion.

*Irrespectively, we have finally decided to publish our model's source code along with the paper, which we believe should now satisfy GMD's code and data policy.*

Regards,

Martina Klose, Oriol Jorba, and Carlos Pérez García-Pando